# 3D-Adapter: Geometry-Consistent Multi-View Diffusion for High-Quality 3D Generation

## Abstract

Multi-view image diffusion models have significantly advanced open-domain 3D object generation. However, most existing models rely on 2D network architectures that lack inherent 3D biases, resulting in compromised geometric consistency. To address this challenge, we introduce 3D-Adapter, a plug-in module designed to infuse 3D geometry awareness into pretrained image diffusion models. Central to our approach is the idea of *3D feedback augmentation*: for each denoising step in the sampling loop, 3D-Adapter decodes intermediate multi-view features into a coherent 3D representation, then re-encodes the rendered RGBD views to augment the pretrained base model through feature addition. We study two variants of 3D-Adapter: a fast feed-forward version based on Gaussian splatting and a versatile training-free version utilizing neural fields and meshes. Our extensive experiments demonstrate that 3D-Adapter not only greatly enhances the geometry quality of text-to-multi-view models such as Instant3D and Zero123++, but also enables high-quality 3D generation using the plain text-to-image Stable Diffusion. Furthermore, we showcase the broad application potential of 3D-Adapter by presenting high quality results in text-to-3D, image-to-3D, text-to-texture, and text-to-avatar tasks.

## 1 Introduction

Diffusion models (Ho et al., 2020; Song et al., 2021) have recently made significant strides in visual synthesis, achieving production-quality results in image generation (Rombach et al., 2022). However, the success of 2D diffusion does not easily translate to the 3D domain due to the scarcity of large-scale datasets and the lack of a unified, neural-network-friendly representation (Po et al., 2024). To bridge the gap between 2D and 3D generation, novel-view or multi-view diffusion models (Liu et al., 2023b; Long et al., 2024; Shi et al., 2023; Li et al., 2024; Chen et al., 2024; Voleti et al., 2024) have been finetuned from pretrained image or video models, facilitating 3D generation via a 2-stage paradigm involving multi-view generation followed by 3D reconstruction (Liu et al., 2023a; 2024a; Li et al., 2024; Wang et al., 2024a; Xu et al., 2024b). While these models generally exhibit good *global semantic consistency* across different view angles, a pivotal challenge lies in achieving *local geometry consistency*. This entails ensuring precise 2D–3D alignment of local features and maintaining geometric plausibility. Consequently, these two-stage methods often suffer from floating artifacts or produce blurry, less detailed 3D outputs (Fig. 1 (c)).

To enhance local geometry consistency, previous works have explored inserting 3D representations and rendering operations into the denoising sampling loop, synchronizing either the denoised outputs (Gu et al., 2023; Xu et al., 2024c; Zuo et al., 2024; Zhang et al., 2024a; Tang et al., 2024c) or the noisy inputs (Liu et al., 2023c; Gao et al., 2024) of the network, a process we refer to as *I/O sync*. However, we observe that I/O sync generally leads to less detailed, overly smoothed textures and geometry (Fig. 1 (b)). This phenomenon can be attributed to two factors:

- Diffusion model sampling is sensitive to error accumulations (Li & van der Schaar, 2024). I/O sync methods insert 3D reconstruction and rendering operations into the denoiser in a way that disrupts the original model topology and introduces errors during each denoising step (unless reconstruction and rendering are perfect).

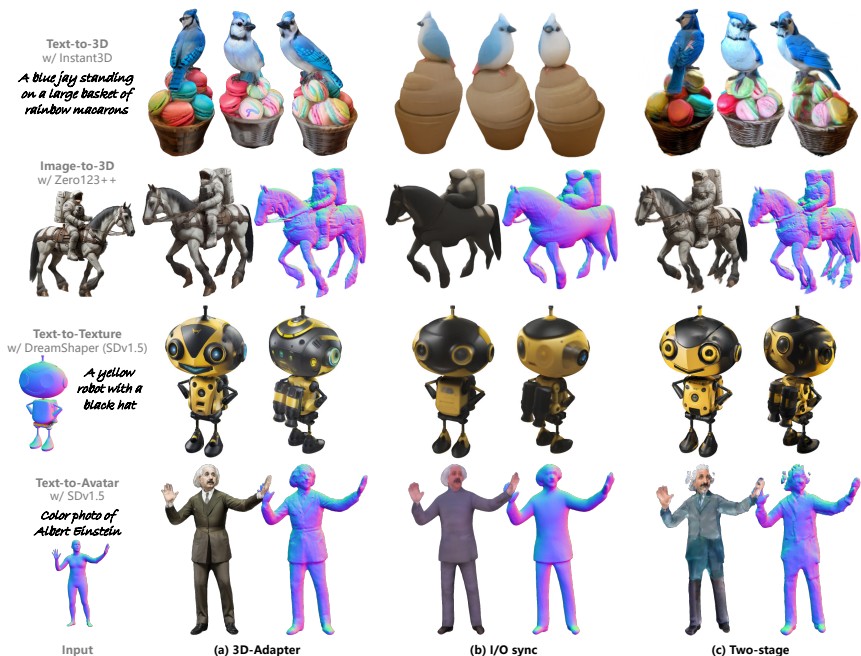

Figure 1: Comparison between the results generated by different architectures. Texture refinement is enabled for text-to-3D, image-to-3D, and text-to-avatar.

- For texture generation methods in Liu et al. (2023c); Gao et al. (2024); Zhang et al. (2024a), I/O sync is equivalent to multi-view score averaging, which theoretically leads to mode collapse, causing the loss of fine details in the generated outputs (analyzed in Appendix A.1).

To overcome the limitations of I/O sync, we propose a novel approach termed *3D feedback augmentation*, which attaches a 3D-aware parallel branch to the base model, while preserving the original network topology and avoiding score averaging. Essentially, this branch decodes intermediate features from the base model to reconstruct an intermediate 3D representation, which is then rendered, encoded, and fed back into the base model through feature addition, thus augmenting 3D awareness. Specifically, when using a denoising U-Net as the base model, we implement 3D feedback augmentation as *3D-Adapter*, which reuses a copy of the original U-Net with an additional 3D reconstruction module to build the parallel branch. Thanks to its ControlNet-like (Zhang et al., 2023) model reuse, 3D-Adapter requires minimal or, in cases where suitable off-the-shelf ControlNets are available, zero training.

To thoroughly evaluate its performance and demonstrate its flexibility, we have tested multiple variants of 3D-Adapter using various base models and reconstruction methods. The base models include Instant3D (Li et al., 2024), Zero123++ (Shi et al., 2023), Stable Diffusion(Rombach et al., 2022) v1.5 and its customizations. The reconstruction methods include GRM (Xu et al., 2024b), texture backprojection, Instant-NGP neural radiance field (NeRF) (Müller et al., 2022; Mildenhall et al., 2020) and DMTet mesh(Shen et al., 2021) optimization. This wide range of possible combinations makes 3D-Adapter models capable of many applications, as shown in Fig. 1. Extensive evaluations show that 3D-Adapter improves geometry consistency compared to the two-stage methods, without suffering from the quality degradation observed with I/O sync.

We summarize the main contributions of this paper as follows:

- We propose 3D-Adapter, which enables high-quality 3D generation with enhanced multi-view geometry consistency by integrating a 3D feedback module into a base image diffusion model.
- We demonstrate that 3D-Adapter is compatible with various base models and reconstruction methods, making it highly adaptable to a range of tasks.
- We conduct extensive experiments to show that 3D-Adapter improves geometry consistency while preserving visual quality, outperforming previous methods on text-to-3D, image-to-3D, and text-to-texture tasks.

## 2 RELATED WORK

**3D-native diffusion models.**    We define 3D-native diffusion models as injecting noise directly into the 3D representations (or their latents) during the diffusion process. Early works (Bautista et al., 2022; Dupont et al., 2022) have explored training diffusion models on low-dimensional latent vectors of 3D representations, but are highly limited in model capacity. A more expressive approach is training diffusion models on triplane representations (Chan et al., 2022), which works reasonably well on closed-domain data (Chen et al., 2023b; Shue et al., 2023; Gupta et al., 2023; Wang et al., 2023). Directly working on 3D grid representations is more challenging due to the cubic computation cost (Müller et al., 2023), so an improved multi-stage sparse volume diffusion model is proposed in Zheng et al. (2023). In general, 3D-native diffusion models face the challenge of limited data, and sometimes the extra cost of preprocessing the training data into 3D representations (e.g., NeRF), which limit their scalability.

**Novel-/multi-view diffusion models.**    Trained on multi-view images of 3D scenes, view diffusion models inject noise into the images (or their latents) and thus benefit from existing 2D diffusion research. Watson et al. (2023) have demonstrated the feasibility of training a conditioned novel view generative model using purely 2D architectures. Subsequent works (Shi et al., 2024; 2023; Liu et al., 2023b; Long et al., 2024; Zheng & Vedaldi, 2024) achieve open-domain novel-/multi-view generation by fine-tuning the pre-trained 2D Stable Diffusion model (Rombach et al., 2022). However, 3D consistency in these models is generally limited to global semantic consistency because it is learned solely from data, without any inherent architectural bias to support detailed local alignment. To this end, Huang et al. (2024b); Kant et al. (2024) have introduced epipolar attention, and Xie et al. (2024) propose to finetune the multi-view model using reinforcement learning.

**Two-stage 3D generation.**    Two-stage methods (Fig. 2 (a)) link view diffusion with multi-view 3D reconstruction models, offering a significant speed advantage over score distillation sampling (SDS) (Poole et al., 2023). Liu et al. (2023a) initially combine Zero-1-to-3 (Liu et al., 2023b) with SparseNeuS (Long et al., 2022), and subsequent works (Liu et al., 2024a; Xu et al., 2024b; Long et al., 2024; Tang et al., 2024a; Hong et al., 2024b; Xu et al., 2024a; Li et al., 2024; Wang et al., 2024b; Yang et al., 2024) have further explored more effective multi-view diffusion models and enhanced reconstruction methods. A common issue with two-stage approaches is that existing reconstruction methods, often designed for or trained under conditions of perfect consistency, lack robustness to local geometric inconsistencies. This may result in floaters and texture seams. To enhance 3D consistency, IM3D (Melas-Kyriazi et al., 2024) applies repeated SDEdit-like refinements (Meng et al., 2022) to the rendered views, which is an orthogonal contribution to our 3D-Adapter.

**View diffusion with 3D representation.**    To introduce 3D representation in single-image diffusion models, Anciukevicius et al. (2023); Tewari et al. (2023) elevate image features into 3D NeRFs to render denoised views. Xu et al. (2024c); Tang et al. (2024c); Zuo et al. (2024) further extend this concept to multi-view diffusion. However, these methods often produce slightly blurry outputs due to error accumulation. Liu et al. (2024b) attempt to preserve the original model topology through attention-based feature fusion, yet it lacks a robust architecture, leading to subpar quality as noted in Liu et al. (2023a). On the other hand, optimization-based I/O sync methods in Gu et al. (2023); Liu et al. (2023c); Gao et al. (2024) either require strong local conditioning or suffer from the pitfalls of score averaging, resulting in overly smoothed textures. Ouroboros3D (Wen et al., 2024) is a concurrent work proposing a similar 3D feedback mechanism to ours. The high-level difference is that Ouroboros3D feeds the rendering to the next denoising timestep, while ours operates within the current timestep.

## 3 PRELIMINARIES

Let $p(\boldsymbol{x}|\boldsymbol{c})$ denote the real data distribution, where $\boldsymbol{c}$ is the condition (e.g., text prompts) and $\boldsymbol{x} \in \mathbb{R}^{V \times 3 \times H \times W}$ denotes the $V$-view images of a 3D object. A Gaussian diffusion model defines a diffusion process that progressively perturb the data point by adding an increasing amount of Gaussian noise $\boldsymbol{\epsilon} \sim \mathcal{N}(0, \boldsymbol{I})$, yielding the noisy data point $\boldsymbol{x}_t := \alpha_t \boldsymbol{x}_i + \sigma_t \boldsymbol{\epsilon}$ at diffusion timestep $t$, with pre-defined noise schedule scalars $\alpha_t, \sigma_t$. A denoising network $D$ is then tasked with removing the noise from $\boldsymbol{x}_t$ to predict the denoised data point. The network is typically trained with an L2

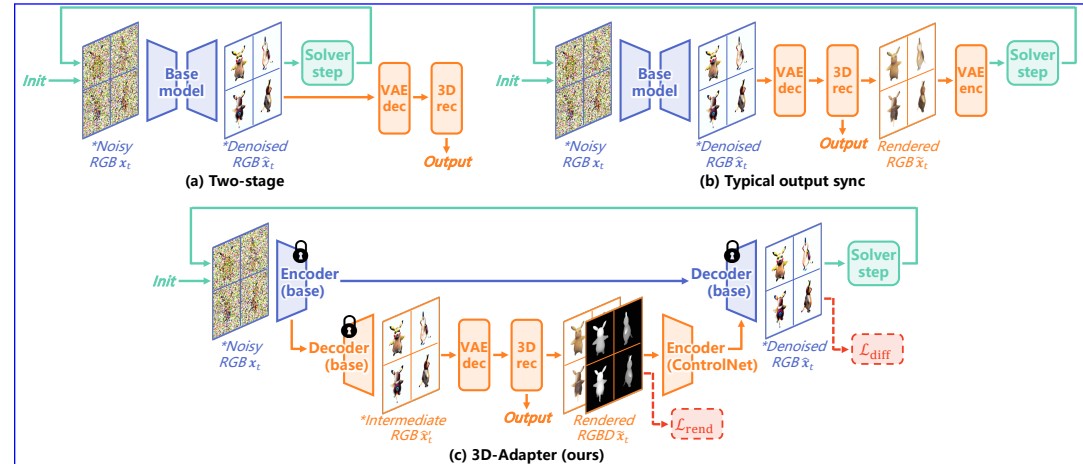

Figure 2: Comparison between different architectures. For brevity, we omit the condition encoders (e.g., text encoders), the rendered alpha channel, and the noisy RGB input for the ControlNet. For LDMs Rombach et al. (2022), VAE encoders and decoders are required, and * denotes RGB latents.

denoising loss:

$$\mathcal{L}_{\text{diff}} = \mathbb{E}_{t,\boldsymbol{c},\boldsymbol{x},\boldsymbol{\epsilon}} \left[ \frac{1}{2} w_t^{\text{diff}} \| D(\boldsymbol{x}_t, \boldsymbol{c}, t) - \boldsymbol{x} \|^2 \right], \tag{1}$$

where $t \sim \mathcal{U}(0, T)$, and $w_t^{\text{diff}}$ is an empirical time-dependent weighting function (e.g., SNR weighting $w_t^{\text{diff}} = (\alpha_t/\sigma_t)^2$). At inference time, one can sample from the model using efficient ODE/SDE solvers (Lu et al., 2022) that recursively denoise $\boldsymbol{x}_t$, starting from an initial noisy state $\boldsymbol{x}_{t_{\text{init}}}$, until reaching the denoised state $\boldsymbol{x}_0$. Note that in latent diffusion models (LDM) (Rombach et al., 2022), both diffusion and denoising occur in the latent space. For brevity, we do not differentiate between latents and images in the equations, assuming VAE encoding and decoding as necessary.

**I/O sync baseline.** We broadly define I/O sync as inserting a 3D representation and a rendering/projecting operation at the input or output end of the denoising network to synchronize multiple views. Input sync is primarily used for texture generation, and it is essentially equivalent to output sync, assuming linearity and synchronized initialization (detailed in the Appendix A.1). Therefore, for simplicity, this paper considers only output sync as the baseline. As depicted in Fig. 2 (b), a typical output sync model can be implemented by reconstructing a 3D representation from the denoised outputs $\hat{\boldsymbol{x}}_t$, and then re-rendering the views from 3D to replace the original outputs.

## 4 3D-ADAPTER

To overcome the limitations of I/O sync, our key idea is the 3D feedback augmentation architecture, which involves reconstructing a 3D representation midway through the denoising network and feeding the rendered views back into the network using ControlNet-like feature addition. This architecture preserves the original flow of the base model while effectively leveraging its inherent priors.

Based on this idea, we propose the 3D-Adapter, as illustrated in Fig. 2 (c). For each denoising step, after passing the input noisy views $\boldsymbol{x}_t$ through the base U-Net encoder, we use a copy of the base U-Net decoder to first output intermediate denoised views $\hat{\boldsymbol{x}}_t'$. A 3D reconstruction model then lifts these intermediate views to a coherent 3D representation, from which consistent RGBD views $\tilde{\boldsymbol{x}}_t$ are rendered and fed back into the network through a ControlNet encoder. The output features from this encoder are added to the base encoder features, which are then processed again by the base decoder to produce the final denoised output $\hat{\boldsymbol{x}}_t$. The full denoising step can be written as:

$$\hat{\boldsymbol{x}}_t = D_{\text{aug}}(\boldsymbol{x}_t, \boldsymbol{c}, t, \underbrace{R(\overbrace{D(\boldsymbol{x}_t, \boldsymbol{c}, t)}^{\hat{\boldsymbol{x}}_t'})}_{\tilde{\boldsymbol{x}}_t}). \tag{2}$$

where $R$ denotes 3D reconstruction and rendering, and $D_{\mathrm{aug}}$ denotes the augmented U-Net with feedback ControlNet.

Various 3D-Adapters can be implemented depending on the choice of base model and 3D reconstruction method, as described in the following subsections.

### 4.1 3D-ADAPTER USING FEED-FORWARD GRM

GRM (Xu et al., 2024b) is a feed-forward sparse-view 3D reconstruction model based on 3DGS. In this section, we describe the method to train GRM-based 3D-Adapters for the text-to-multi-view model Instant3D (Li et al., 2024) and image-to-multi-view model Zero123++ (Shi et al., 2023).

**Training phase 1: finetuning GRM.** GRM is originally trained on consistent ground truth input views, and is not robust to low-quality intermediate views, which are often highly inconsistent and blurry. To overcome this challenge, we first finetune GRM using the intermediate images $\hat{x}_t'$ as inputs, where the time $t$ is randomly sampled just like in the diffusion loss. In this training phase, we freeze the base encoder and decoder of the U-Net, and initialize GRM with the official checkpoint for finetuning. As shown in Fig. 2 (c), a rendering loss $\mathcal{L}_{\mathrm{rend}}$ is employed to supervise GRM with ground truth novel views. Specifically, both the appearance and geometry are supervised using the combination of an L1 loss $L_1^{\mathrm{RGBAD}}$ on RGB/alpha/depth maps, and an LPIPS loss $L_{\mathrm{LPIPS}}^{\mathrm{RGB}}$ (Zhang et al., 2018) on RGB only. The loss is computed on 16 rendered views $\tilde{X}_t \in \mathbb{R}^{16 \times 5 \times 512 \times 512}$ and the corresponding ground truth views $X_{\mathrm{gt}}$, given by:

$$\mathcal{L}_{\mathrm{rend}} = \mathbb{E}_{t,c,x,\epsilon}\left[ w_t^{\mathrm{rend}}\left( L_1^{\mathrm{RGBAD}}\left( \tilde{X}_t, X_{\mathrm{gt}} \right) + L_{\mathrm{LPIPS}}^{\mathrm{RGB}}\left( \tilde{X}_t, X_{\mathrm{gt}} \right) \right) \right], \tag{3}$$

where $w_t^{\mathrm{rend}}$ is a time-dependent weighting function. We use $w_t^{\mathrm{rend}} = \alpha_t/\sqrt{\alpha_t^2 + \sigma_t^2}$. The L1 RGBAD loss also employs channel-wise weights, which are detailed in our code.

**Training phase 2: finetuning feedback ControlNet.** In this training phase, we freeze all modules except the feedback ControlNet encoder, which is initialized with the base U-Net weights for finetuning. Following standard ControlNet training method, we employ the diffusion loss in Eq. (1) to finetune the RGBD feedback ControlNet. To accelerate convergence, we feed rendered RGBD views of a less noisy timestep $\tilde{x}_{0.1t}$ to the ControlNet during training.

**Inference: guided 3D feedback augmentation.** One potential issue is that the ControlNet encoder may overfit the finetuning dataset, resulting in an undesirable bias that persists even if the rendered RGBD $\tilde{x}_t$ is replaced with a zero tensor. To mitigate this issue, inspired by classifier-free guidance (CFG) (Ho & Salimans, 2021), we replace $\hat{x}_t$ with the guided denoised views $\hat{x}_t^{\mathrm{G}}$ during inference to cancel out the ControlNet bias:

$$\hat{x}_t^{\mathrm{G}} = \lambda_{\mathrm{aug}}(D_{\mathrm{aug}}(x_t, c, t, \tilde{x}_t) - D_{\mathrm{aug}}(x_t, c, t, \mathbf{0})) + \lambda_c D(x_t, c, t) + (1 - \lambda_c)D(x_t, \mathbf{0}, t), \tag{4}$$

where $\lambda_c$ is the regular condition CFG scale, and $\lambda_{\mathrm{aug}}$ is our feedback augmentation guidance scale. During training, we feed zero tensors to the ControlNet with a 20% probability, so that $D_{\mathrm{aug}}(x_t, c, t, \mathbf{0})$ learns a meaningful dataset bias.

**Training details.** We adopt various techniques to reduce the memory footprint, including mixed precision training, 8-bit AdamW (Dettmers et al., 2022; Loshchilov & Hutter, 2019), gradient checkpointing, and deferred back-propagation (Xu et al., 2024b; Zhang et al., 2022). The adapter is trained with a total batch size of 16 objects on 4 A6000 GPUs (VRAM usage peaks at 39GB). In phase 1, GRM is finetuned with a small learning rate of $5 \times 10^{-6}$ for 2k iterations (for Instant3D, taking 3 hours) or 4k iterations (for Zero123++, taking 9 hours). In phase 2, ControlNet is finetuned with a learning rate of $1 \times 10^{-5}$ for 5k iterations (taking 8 hours for Instant3D and 5 hours for Zero123++).

47k (for Instant3D) or 80k (for Zero123++) objects from a high-quality subset of Objaverse (Deitke et al., 2023) are rendered as the training data.

### 4.2 3D-ADAPTER USING 3D OPTIMIZATION/TEXTURE BACKPROJECTION

Feed-forward 3D reconstruction methods, like GRM, are typically constrained by specific camera layouts. In contrast, more flexible reconstruction approaches, such as optimizing a NeRF (Mildenhall et al., 2020) or mesh, can accommodate diverse camera configurations and achieve higher-quality results with denser cameras, although they require longer optimization times.

To demonstrate the compatibility with optimization-based reconstruction methods, we explore a new variation of 3D-Adapter (Fig. 9), using Instant-NGP NeRF (Müller et al., 2022) and DMTet mesh (Shen et al., 2021) optimization as the reconstruction module, with Stable Diffusion v1.5 (Rombach et al., 2022) being the base model. For feedback augmentation, Stable Diffusion comes with off-the-shelf ControlNets (Zhang et al., 2023), which empirically work very well as the feedback encoder. Specifically, we simultaneously use the "tile" ControlNet (originally trained for superresolution) for RGB feedback, and the depth ControlNet for depth feedback. Dense cameras are randomly generated around the object for multi-view diffusion. Since Stable Diffusion is a single image model, 3D-Adapter (or I/O sync) is the only module that synchronizes multi-view samples.

Alternatively, for texture generation only (Section 5.4), multi-view aggregation can be achieved by backprojecting the views into UV space and blending the results according to visibility.

**Details on NeRF/mesh optimization.** During the sampling process, the adapter performs NeRF optimization for the first 60% of the denoising steps. It then converts the color and density fields into a texture field and DMTet mesh, respectively, to complete the remaining 40% denoising steps. All optimizations are incremental, meaning the 3D state from the previous denoising step is retained to initialize the next. As a result, only 96 optimization steps are needed per denoising step. we employ L1 and LPIPS losses on RGB and alpha maps, and total variation (TV) loss on normal maps. Additionally, we enforce stronger geometry regularization using ray entropy loss for NeRF, and Laplacian smoothing loss (Sorkine et al., 2004) plus normal consistency loss for mesh, making the optimization more robust to imperfect intermediate views $\hat{x}'_t$. More details can be found in Appendix C.

**Limitations.** It should be noted that, when using single-image diffusion as the base model, 3D-Adapter alone cannot provide the necessary global semantic consistency for 3D generation. Therefore, it should be complemented with other sources of consistency, such initialization with partial noise like SDEdit (Meng et al., 2022) or extra conditioning from ControlNets. For text-to-avatar generation, we use rendered views of a human template for SDEdit initialization with the initial timestep $t_{init}$ set to $0.88T$, and employ an extra pose ControlNet for conditioning. For text-to-texture generation, global consistency is usually good due to ground truth depth conditioning.

## 4.3 TEXTURE POST-PROCESSING

To further enhance the visual quality of objects generated from text, we implement an optional texture refinement pipeline as a post-processing step. First, when using the GRM-based 3D-Adapter, we convert the generated 3DGS into a textured mesh via TSDF integration. With the initial mesh, we render six surrounding views and apply per-view SDEdit refinement ($t_{init} = 0.5T$) using Stable Diffusion v1.5 with "tile" ControlNet. Finally, the refined views are aggregated into the UV space using texture backprojection. For fair comparisons in the experiments, this refinement step is not used by default unless specified otherwise.

## 5 EXPERIMENTS

## 5.1 EVALUATION METRICS

To evaluate the results generated by 3D-Adapter and compare them to various baselines and competitors, we compute the following metrics based on the rendered images of the generated 3D representations:

- **CLIP score** (Radford et al., 2021; Jain et al., 2022): Evaluates image–text alignment in text-to-3D, text-to-texture, and text-to-avatar tasks. We use CLIP-ViT-L-14 for all CLIP-related metrics.
- **Aesthetic score** (Schuhmann et al., 2022): Assesses texture details. The user study in Wu et al. (2024) revealed that this metric highly correlates with human preference in texture details.
- **FID** (Heusel et al., 2017): Measures the visual quality when reference test set images are available, applicable to text-to-3D models trained on common dataset and all image-to-3D models.
- **CLIP similarity** (Radford et al., 2021), **LPIPS** (Zhang et al., 2018), **SSIM** (Wang et al., 2004), **PSNR**: Evaluates novel view fidelity in image-to-3D.

Table 1: Text-to-3D: comparison with baselines, parameter sweep, and ablation studies, on the validation set.

| ID | Method | CLIP↑ | Aesthetic↑ | FID↓ | MDD /$10^{-7}$↓ |
|---|---|---|---|---|---|
| A0 | Two-stage (original GRM) | 27.02 | 4.48 | 34.19 | 232.4 |
| A1 | I/O sync (original GRM) | 24.62 | 4.35 | 63.22 | 1239.7 |
| A2 | A1 + GRM finetuning | 22.57 | 4.16 | 70.35 | **1.7** |
| A3 | A2 + dynamic blending | 25.95 | 4.39 | 44.62 | 2.8 |
| B0 | 3D-Adapter $\lambda_{aug}$=1 | **27.31** | **4.54** | **32.81** | 4.7 |
| B1 | 3D-Adapter $\lambda_{aug}$=2 | 27.22 | 4.52 | 33.46 | 3.9 |
| B2 | 3D-Adapter $\lambda_{aug}$=4 | 26.99 | 4.45 | 34.34 | 3.2 |
| B3 | 3D-Adapter $\lambda_{aug}$=8 | 25.47 | 4.28 | 39.36 | 25.3 |
| C0 | B0 w/o feedback | 27.18 | **4.55** | 33.13 | 7.6 |
| C1 | B0 w/o bias canceling | 25.49 | 4.36 | 42.20 | 3.6 |

Table 2: Text-to-3D: comparison with previous SOTAs.

| Method | Type | CLIP↑ | Aesthetic↑ | Time↓ |
|---|---|---|---|---|
| Shap-E | Mesh | 19.4 | 4.07 | 9 s |
| 3DTopia | Mesh | 21.2 | 4.40 | 3 m |
| LGM | GS | 22.5 | 4.31 | **5 s** |
| Instant3D | NeRF | 25.5 | 4.24 | 20 s |
| MVDream-SDS | NeRF | 26.9 | 4.49 | 1 h |
| GRM | GS | 26.6 | 4.54 | 8 s |
| 3D-Adapter (ours) | GS | 27.7 | 4.61 | 23 s |
| 3D-Adapter + tex refine (ours) | Mesh | **28.0** | **4.71** | 1 m |

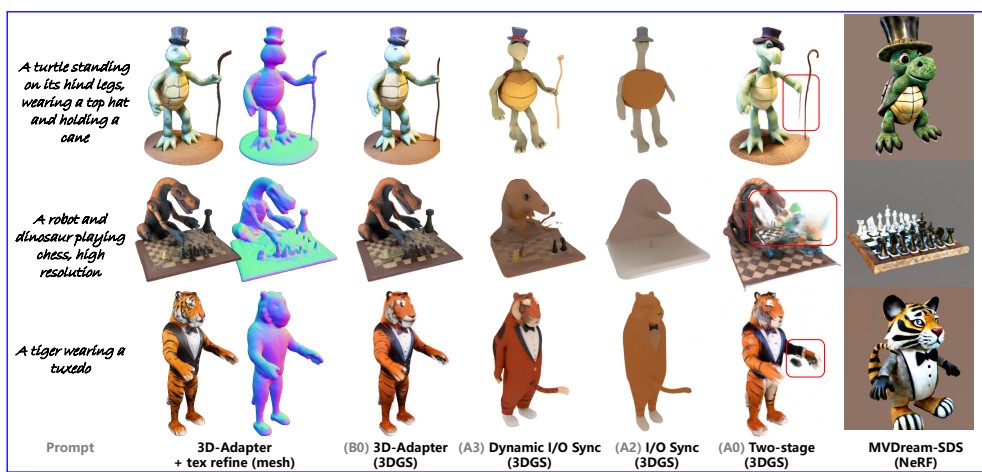

Figure 3: Comparison on text-to-3D generation. Both 3D-Adapter and I/O sync fix the broken geometry and floaters present in the two-stage method, but I/O sync suffers from blurriness.

- **Mean depth distortion (MDD)** (Yu et al., 2024; Huang et al., 2024a): Assesses the geometric quality of generated 3DGS. Lower depth distortion indicates less floaters or fuzzy surfaces, reflecting better geometry consistency. More details can be found in Appendix D.
- **CLIP t-less score**: Assesses the geometric quality of generated meshes by computing the CLIP score between shaded textureless renderings and texts appended with "textureless 3D model".

Additionally, we report inference times measured on a single RTX 6000 GPU, with file system I/O and UV unwrapping (if applicable) included.

## 5.2 TEXT-TO-3D GENERATION

For text-to-3D generation, we adopt the GRM-based 3D-Adapter with Instant3D U-Net as the base model. All results are generated using EDM Euler ancestral solver (Karras et al., 2022) with 30 denoising steps and mean latent initialization (Appendix B.2). The inference time is around 0.7 sec per step, and detailed inference time analysis is presented in Appendix B.3. For evaluation, we first compare 3D-Adapter with the baselines and conduct ablation studies on a validation set of 379 BLIP-captioned objects sampled from a high-quality subset of Objaverse (Li et al., 2022; Deitke et al., 2023). The results are shown in Table 1, with the rendered images from the validation set used as real samples when computing the FID metric. Subsequently, we benchmark 3D-Adapter on the same test set as GRM (Xu et al., 2024b), consisting of 200 prompts, to make fair comparisons to the previous SOTAs in Table 2. Qualitative results are shown in Fig. 3.

**Baselines.** The two-stage GRM (A0) exhibits good visual quality, but the MDD metric is magnitudes higher than that of our 3D-Adapter (B0–B3) due to the highly ambiguous geometry caused by local misalignment. Naively rewiring it into an I/O sync model (A1) worsens the results, as the original GRM is trained only on rendered ground truths $x$ and cannot handle the imperfections of

Table 3: Image-to-3D: comparison with previous SOTAs.

| Method | Type | PSNR↑ | SSIM↑ | LPIPS↓ | CLIP sim ↑ | FID↓ | Time↓ |
|---|---|---|---|---|---|---|---|
| One-2-3-45 | Mesh | 17.84 | 0.800 | 0.199 | 0.832 | 89.4 | 45 s |
| TriplaneGaussian | GS | 16.81 | 0.797 | 0.257 | 0.840 | 52.6 | **0.2 s** |
| Shap-E | Mesh | 15.45 | 0.772 | 0.297 | 0.854 | 56.5 | 9 s |
| LGM | GS | 16.90 | 0.819 | 0.235 | 0.855 | 42.1 | 5 s |
| EpiDiff-GRM | GS | 18.52 | 0.806 | 0.244 | 0.859 | 61.1 | 55 s |
| DreamGaussian | Mesh | 19.19 | 0.811 | 0.171 | 0.862 | 57.6 | 2 m |
| Wonder3D | Mesh | 17.29 | 0.815 | 0.240 | 0.871 | 55.7 | 3 m |
| CRM | Mesh | 18.04 | 0.809 | 0.217 | 0.871 | 61.9 | 13 s |
| One-2-3-45++ | Mesh | 17.79 | 0.819 | 0.219 | 0.886 | 42.1 | 1 m |
| InstantMesh | Mesh | 19.24 | 0.828 | 0.156 | 0.921 | 25.6 | 32 s |
| GRM | GS | 20.10 | 0.826 | 0.136 | 0.932 | 27.4 | 6 s |
| 3D-Adapter (ours) | GS | **20.38** | **0.840** | **0.135** | **0.936** | **20.2** | 23 s |
| 3D-Adapter + TSDF (ours) | Mesh | 20.34 | **0.840** | **0.135** | 0.933 | 21.7 | 35 s |

Table 4: Text-to-avatar: comparison with baselines.

| Methods | CLIP↑ | Aesthetic↑ | CLIP t-less↑ |
|---|---|---|---|
| Two-stage baseline | 23.90 | 4.79 | 24.60 |
| I/O sync baseline | 22.01 | 4.53 | 25.98 |
| 3D-Adapter | 23.67 | 4.97 | **26.07** |
| 3D-Adapter + tex refine | **24.07** | **5.11** | **26.07** |

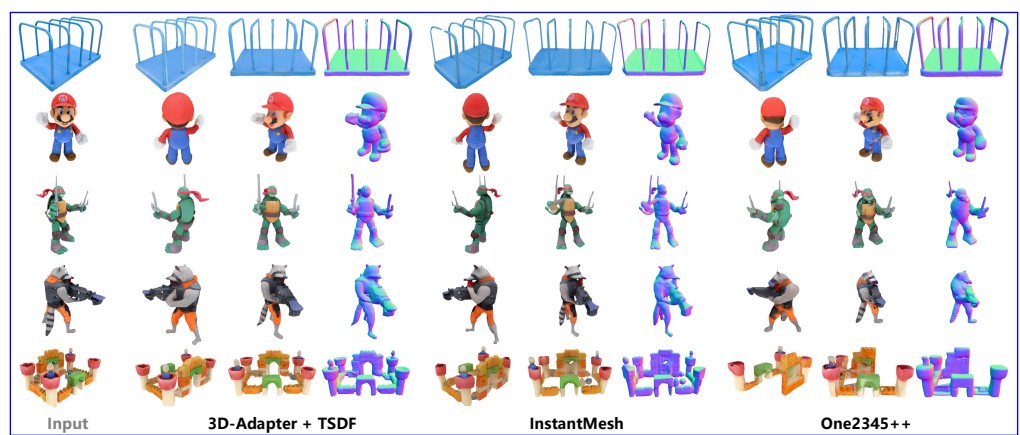

Figure 4: Comparison of mesh-based image-to-3D methods on the GSO test set.

the denoised views $\hat{x}$. When using the GRM model fine-tuned according to our method (Eq. 3), the model (A2) achieves the lowest possible MDD with nearly perfect geometry consistency, which validates the effectiveness of our GRM finetuning approach. However, it suffers significantly from mode collapse and yields the worst appearance metrics (analyzed in Appendix A.2). Adopting a dynamic blending technique (Appendix A.2) for I/O sync (A3) alleviates this issue, but the appearance metrics are still worse than two-stage (A0).

**Parameter sweep on $\lambda_{\mathrm{aug}}$ and ablation studies.** The 3D-Adapter with a feedback augmentation guidance scale $\lambda_{\mathrm{aug}} = 1$ (B0) achieves the best visual quality among all variants and significantly better geometry quality than A0. As $\lambda_{\mathrm{aug}}$ increases, the MDD metric continues to improve, but at the expense of visual quality. A very large $\lambda_{\mathrm{aug}}$ (B3) unsurprisingly worsens the results, similar to a large CFG scale. Disabling feedback augmentation (C0, equivalent to $\lambda_{\mathrm{aug}} = 0$) notably impacts the geometric quality, as evidenced by the worse MDD metric, although it still outperforms the baseline (A0) thanks to our robust GRM fine-tuning. Additionally, we ablated the bias canceling technique (C1), observing significant degradation in all visual metrics, which substantiates the effectiveness of 3D feedback guidance (Eq. (4)). Qualitative results are all presented in Fig. 10.

**Comparison with other competitors.** Built on top of GRM, our 3D-Adapter ($\lambda_{\mathrm{aug}}=1$) further advances the benchmark, outperforming previous SOTAs in text-to-3D (Jun & Nichol, 2023; Tang et al., 2024a; Li et al., 2024; Shi et al., 2024; Xu et al., 2024b; Hong et al., 2024a) as shown in Table 2 and Fig. 3.

### 5.3 IMAGE-TO-3D GENERATION

For image-to-3D generation, we adopt the same approach used for text-to-3D generation, except for employing Zero123++ U-Net as the base model and using 40 denoising steps. We follow the same evaluation protocol as in Xu et al. (2024b), using 248 GSO objects (Downs et al., 2022) as

Table 5: Text-to-texture: comparison with baselines.

| Methods | CLIP↑ | Aesthetic↑ |
|---|---|---|
| Two-stage baseline | 25.82 | **4.85** |
| I/O sync baseline | 26.05 | 4.68 |
| 3D-Adapter + I/O sync | **26.41** | 4.61 |
| 3D-Adapter | 26.40 | **4.85** |

Table 6: Text-to-texture: comparison with previous SOTAs.

| Methods | CLIP↑ | Aesthetic↑ | Time↓ |
|---|---|---|---|
| TexPainter | 25.36 | 4.55 | 11.6 s |
| TEXTure | 25.39 | 4.66 | 2.0 m |
| Text2Tex | 24.44 | 4.72 | 11.2 m |
| SyncMVD | 25.65 | 4.76 | 1.9 m |
| 3D-Adapter (ours) | **26.40** | **4.85** | **1.5 m** |

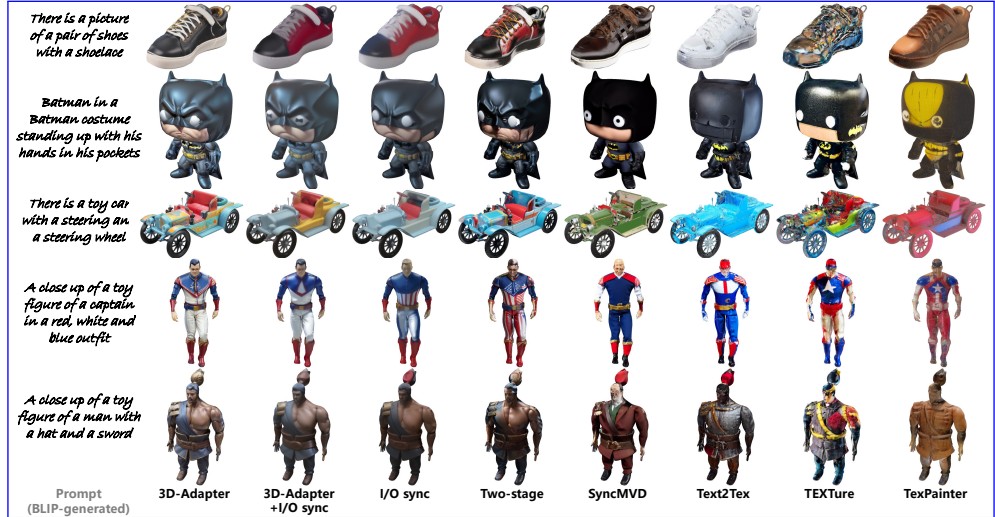

Figure 5: Comparison on text-to-texture generation.

the test set. As shown in Table 3, 3D-Adapter ($\lambda_{\text{aug}}$=1) outperforms the two-stage GRM and other competitors (Xu et al., 2024a; Wang et al., 2024b; Liu et al., 2023a; Zou et al., 2024; Jun & Nichol, 2023; Tang et al., 2024a;b; Long et al., 2024; Liu et al., 2024a; Xu et al., 2024b; Huang et al., 2024b) on all metrics. Moreover, the quality loss in converting the generated 3DGS to mesh via TSDF is almost negligible. We present qualitative comparisons of the meshes generated by 3D-Adapter and other methods in Fig. 4.

## 5.4 TEXT-TO-TEXTURE GENERATION

For text-to-texture evaluation, 3D-Adapter employs fast texture backprojection to blend multiple views for intermediate timesteps, and switches to high-quality texture field optimization (similar to NeRF) for the final timestep. A community Stable Diffusion v1.5 variant, DreamShaper 8, is adopted as the base model. During the sampling process, 32 surrounding views are used initially, and this number is gradually reduced to 7 views during the denoising process to reduce computation in later stages. We adopt the EDM Euler ancestral solver with 24 denoising steps. 92 BLIP-captioned objects are sampled from a high-quality subset of Objaverse as our test set.

**Comparison with baselines.** As shown in Table 5 and Fig. 5, the two-stage baseline has good texture details but notably worse CLIP score due to poor consistency. The I/O sync baseline has much better consistency, but it sacrifices details, resulting in the worst aesthetic score. In comparison, 3D-Adapter excels in both metrics, producing detailed and consistent textures. Additionally, we demonstrate that 3D-Adapter and I/O sync should not be used simultaneously, as I/O sync consistently compromises texture details, as evidenced by the Aesthetic score and qualitative results.

**Comparison with other competitors.** We compare 3D-Adapter with SyncMVD (Liu et al., 2023c), Text2Tex (Chen et al., 2023a), TEXTure (Richardson et al., 2023), and TexPainter (Zhang et al., 2024a), where SyncMVD and TexPainter are also I/O sync methods. Quantitatively, Table 6 demonstrates that 3D-Adapter significantly outperforms previous SOTAs on both metrics. Interestingly, even our two-stage baseline in Table 5 surpasses the competitors, which can be attributed to

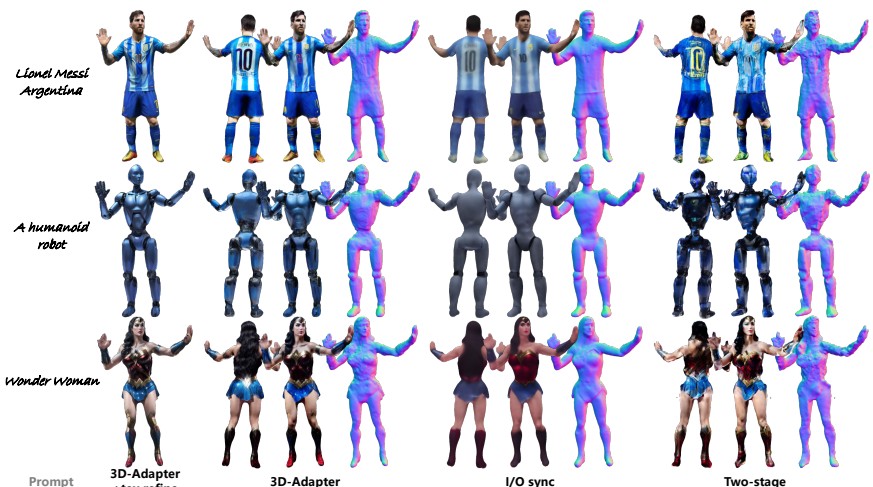

Figure 6: Comparison on text-to-avatar generation using the same pose template.

our use of texture field optimization and community-customized base model. Qualitative results in Fig. 5 reveal that previous methods are generally less robust compared to 3D-Adapter and may produce artifacts in some cases.

**Limitations.** 3D-Adapter and the methods in Table. 6 do not disentangle texture from lighting. PBR texture generation (Zhang et al., 2024b; Zeng et al., 2024; Youwang et al., 2024; Deng et al., 2024) using 3D-Adapter could be a potential future extension of this work.

### 5.5 TEXT-TO-AVATAR GENERATION

For text-to-avatar generation, the optimization-based 3D-Adapter is adopted with a custom pose ControlNet for Stable Diffusion v1.5, which provides extra conditioning given a human pose template. 32 full-body views and 32 upper-body views are selected for denoising, capturing both the overall figure and face details. These are later reduced to 12 views during the denoising process. We use the EDM Euler ancestral solver with 32 denoising steps, with an inference time of approximately 7 minutes per object. Texture editing (using text-to-texture pipeline and SDEdit with $t_{\mathrm{init}} = 0.3T$) and refinement can be optionally applied to further improve texture details, which costs 1.4 minutes. For evaluation, we compare 3D-Adapter with baselines using 21 character prompts on the same pose template. As shown in Table 4, 3D-Adapter achieves the highest scores across all three metrics, indicating superior appearance and geometry. Fig. 6 reveals that I/O sync produces overly smoothed texture and geometry due to mode collapse, while the two-stage baseline results in noisy, less coherent texture and geometry. These observations also align with the quantitative results in Table 4.

## 6 CONCLUSION

In this work, we have introduced 3D-Adapter, a plug-in module that effectively enhances the 3D geometry consistency of existing multi-view diffusion models, bridging the gap between high-quality 2D and 3D content creation. We have demonstrated two variants of 3D-Adapter: the fast 3D-Adapter using feed-forward Gaussian reconstruction, and the flexible training-free 3D-Adapter using 3D optimization and pretrained ControlNets. Experiments on text-to-3D, image-to-3D, text-to-texture, and text-to-avatar tasks have substantiated its all-round competence, suggesting great generality and potential in future extension.

**Limitations.** 3D-Adapter introduces substantial computation overhead, primarily due to the VAE decoding process before 3D reconstruction. In addition, we observe that our finetuned ControlNet for 3D feedback augmentation strongly overfits the finetuning data, which may limit its generalization despite the proposed guidance method. Future work may focus on developing more efficient, easy-to-finetune networks for 3D-Adapter.

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

# A DETAILS ON THE I/O SYNC BASELINE

## A.1 THEORETICAL ANALYSIS

When performing diffusion ODE sampling using the common Euler solver, a linear input sync operation (e.g., linear blending or optimizing using the L2 loss) is equivalent to syncing the output $\hat{\boldsymbol{x}}_t$ as well as the initialization $\boldsymbol{x}_{t_{\text{init}}}$. This is because the input $\boldsymbol{x}_t$ can be expressed as a linear combination of all previous outputs $\{\boldsymbol{x}_{t-\Delta t}, \boldsymbol{x}_{t-2\Delta t}, \dots\}$ and the initialization $\boldsymbol{x}_{t_{\text{init}}}$ by expanding the recursive Euler steps.

Furthermore, linear I/O sync is also equivalent to linear score sync, since the learned score function $\boldsymbol{s}_t(\boldsymbol{x}_t)$ can also be expressed as a linear combination of the input $\boldsymbol{x}_t$ and output $\hat{\boldsymbol{x}}_t$:

$$\boldsymbol{s}_t(\boldsymbol{x}_t) = -\frac{\hat{\boldsymbol{\epsilon}}_t}{\sigma_t} = \frac{\alpha_t \hat{\boldsymbol{x}}_t - \boldsymbol{x}_t}{\sigma_t{}^2} \tag{5}$$

However, synchronizing the score function, a.k.a. score averaging, is theoretically problematic. Let $p(\boldsymbol{x}|\boldsymbol{c}_1), p(\boldsymbol{x}|\boldsymbol{c}_2)$ be two independent probability density functions of a corresponding pixel $\boldsymbol{x}$ viewed from cameras $\boldsymbol{c}_1$ and $\boldsymbol{c}_2$, respectively. A diffusion model is trained to predict the score function $\boldsymbol{s}_t(\boldsymbol{x}_t|\boldsymbol{c}_v)$ of the noisy distribution at timestep $t$, defined as:

$$\boldsymbol{s}_t(\boldsymbol{x}_t|\boldsymbol{c}_v) = \nabla_{\boldsymbol{x}_t} \log \int p(\boldsymbol{x}_t|\boldsymbol{x})p(\boldsymbol{x}|\boldsymbol{c}_v)d\boldsymbol{x}, \tag{6}$$

where $p(\boldsymbol{x}_t|\boldsymbol{x}) = \mathcal{N}(\boldsymbol{x}_t; \alpha_t \boldsymbol{x}, \sigma_t^2 \boldsymbol{I})$ is a Gaussian perturbation kernel. Ideally, assuming $\boldsymbol{c}_1$ and $\boldsymbol{c}_2$ are independent, combining the two conditional PDFs $p(\boldsymbol{x}|\boldsymbol{c}_1)$ and $p(\boldsymbol{x}|\boldsymbol{c}_2)$ yields the product $p(\boldsymbol{x}|\boldsymbol{c}_1, \boldsymbol{c}_2) = \frac{1}{Z}p(\boldsymbol{x}|\boldsymbol{c}_1)p(\boldsymbol{x}|\boldsymbol{c}_2)$, where $Z$ is a normalization factor. The corresponding score function should then become $\boldsymbol{s}_t(\boldsymbol{x}_t|\boldsymbol{c}_1, \boldsymbol{c}_2) = \nabla_{\boldsymbol{x}_t} \log \int p(\boldsymbol{x}_t|\boldsymbol{x})p(\boldsymbol{x}|\boldsymbol{c}_1, \boldsymbol{c}_2)d\boldsymbol{x}$. However, the average of $\boldsymbol{s}_t(\boldsymbol{x}_t|\boldsymbol{c}_1)$ and $\boldsymbol{s}_t(\boldsymbol{x}_t|\boldsymbol{c}_2)$ is generally not proportional to $\boldsymbol{s}_t(\boldsymbol{x}_t|\boldsymbol{c}_1, \boldsymbol{c}_2)$, i.e.:

$$\frac{1}{2}\boldsymbol{s}_t(\boldsymbol{x}_t|\boldsymbol{c}_1) + \frac{1}{2}\boldsymbol{s}_t(\boldsymbol{x}_t|\boldsymbol{c}_2) = \frac{1}{2}\nabla_{\boldsymbol{x}_t} \log \left( \int p(\boldsymbol{x}_t|\boldsymbol{x})p(\boldsymbol{x}|\boldsymbol{c}_1)d\boldsymbol{x} \right)\left( \int p(\boldsymbol{x}_t|\boldsymbol{x})p(\boldsymbol{x}|\boldsymbol{c}_2)d\boldsymbol{x} \right)$$

$$\not\propto \nabla_{\boldsymbol{x}_t} \log \int p(\boldsymbol{x}_t|\boldsymbol{x})\left( \frac{1}{Z}p(\boldsymbol{x}|\boldsymbol{c}_1)p(\boldsymbol{x}|\boldsymbol{c}_2) \right)d\boldsymbol{x} = \boldsymbol{s}_t(\boldsymbol{x}_t|\boldsymbol{c}_1, \boldsymbol{c}_2). \tag{7}$$

In Fig. 7, we illustrate a simple 1D simulation, showing that score averaging leads to mode collapse, when compared to the real product distribution. This explains the blurry, mean-shaped results produced by the I/O sync baselines. This problem is also noted in a concurrent work (Bradley & Nakkiran, 2024) in the context of classifier-free guidance.

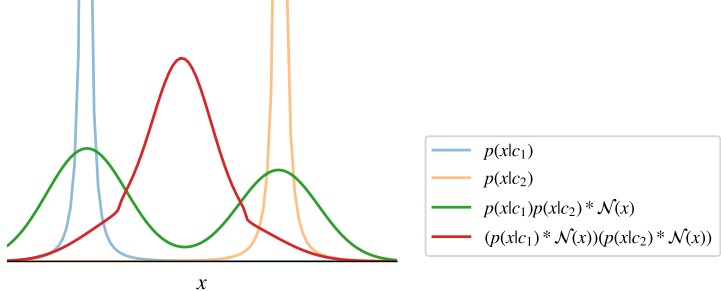

Figure 7: A simple 1D simulation illustrating the difference between the **score averaged distribution** and the actual **perturbed product distribution**. $*$ denotes convolution, and $\mathcal{N}(x)$ denotes the Gaussian perturbation kernel.

## A.2 DYNAMIC I/O SYNC

While I/O sync works reasonably on our texture generation benchmark, our text-to-3D model using I/O sync (A2 in Table 1 and Fig. 3) exhibits significant quality degradation due to mode collapse.

Table 7: GRM-based 3D-Adapter: Inference times (sec) with guidance on a single RTX A6000.

| Encode | Adapter Decode | VAE Decode | GRM | Render | Adapter Encode | Decode | Adapter total | Overall total |
|--------|----------------|------------|------|--------|----------------|--------|---------------|---------------|
| 0.055  | 0.120          | 0.215      | 0.091 | 0.023  | 0.082          | 0.121  | 0.531         | 0.707         |

We believe the main reasons are twofold. First, the base model Instant3D generates a very sparse set of only four views, which are hard to synchronize. Second, our finetuned GRM reconstructor is trained using the depth loss to suppress surface fuzziness, which has a negative impact when its sharp renderings $\tilde{x}_t$ are used as diffusion output. This is because a well-trained diffusion model should actually predict blurry outputs $\hat{x}_t$ in the early denoising stage as the mean of the distribution $p(x_0|x_t)$. Only in the late stage should $\hat{x}_t$ be sharp and crisp, as shown in Fig. 8.

To make the I/O sync baseline more competitive on the text-to-3D benchmark, we adopt a simple technique called *dynamic blending* or dynamic I/O sync. The idea is that, since I/O sync mainly corrupts fine-grained details, its influence should be reduced during the late denoising stages when details are being generated. Therefore, we perform a weighted blending of the denoised views before synchronization $\hat{x}_t$ and the rendered views $\tilde{x}_t$:

$$\tilde{x}_t^{\text{blend}} = (1 - \lambda_t^{\text{sync}})\hat{x}_t + \lambda_t^{\text{sync}}\tilde{x}_t, \tag{8}$$

where $\lambda_t^{\text{sync}}$ is a time-dependent blending weight, and $\tilde{x}_t^{\text{blend}}$ is the blended output that is fed to the diffusion solver. We set $\lambda_t^{\text{sync}} = \frac{1-\alpha_t}{\sqrt{\alpha_t^2 + \sigma_t^2}}$, so that $\lambda_t^{\text{sync}}$ decreases over the denoising process.

As shown in Table 1 and Fig. 3, dynamic I/O sync demonstrates significant improvements in visual quality over vanilla I/O sync. However, its MDD metric becomes worse than vanilla I/O sync, and the visual quality is still clearly below that of the two-stage method and 3D-Adapter. While it is possible to tune a better blending weight $\lambda_t^{\text{sync}}$, we believe it is very difficult to reduce the gap due to the aforementioned challenges brought by our model setup.

## B DETAILS ON GRM-BASED 3D-ADAPTER

### B.1 CONTROLNET

The GRM-based 3D-Adapter trains a ControlNet (Zhang et al., 2023) for feedback augmentation, which has very large model capacity and can easily overfit our relatively small finetuning dataset (e.g., 47k objects for Instant3D). Therefore, using the CFG-like bias subtraction technique (Eq. (4)) is extremely important to the generalization performance, which is already validated in our ablation studies. Additionally, we disconnect the text prompt input from the ControlNet to further alleviate overfitting.

### B.2 MEAN LATENT INITIALIZATION

Instant3D's 4-view UNet is sensitive to the initialization method, as noted in the original paper (Li et al., 2024), which develops an empirical Gaussian blob initialization method to stabilize the background color. In contrast, this paper adopts a more principled mean latent initialization method by computing the mean value $\bar{x}$ of the VAE-encoded latents of 10K objects in the training set. The initial state is then sampled by perturbing the mean latent with Gaussian noise $\epsilon$:

$$x_{t_{\text{init}}} = \alpha_{t_{\text{init}}}\bar{x} + \sigma_{t_{\text{init}}}\epsilon. \tag{9}$$

### B.3 INFERENCE TIME

Detailed module-level inference times per denoising step is shown in Table 7 (with classifier-free guidance and guided 3D feedback augmentation enabled). Apparently, the SDXL VAE decoder is the most expensive module within 3D-Adapter, which may be replaced by a more efficient decoder in future work.

## C  DETAILS ON OPTIMIZATION-BASED 3D-ADAPTER

The optimization-based 3D-Adapter faces the challenge of potentially inconsistent multi-view inputs, especially at the early denoising stage. Existing surface optimization approaches, such as NeuS (Wang et al., 2021a), are not designed to address the inconsistency. Therefore, we have developed various techniques for the robust optimization of InstantNGP NeRF (Müller et al., 2022) and DMTet mesh (Shen et al., 2021), using enhanced regularization and progressive resolution.

**Rendering.** For each NeRF optimization iteration, we randomly sample a $128 \times 128$ image patch from all camera views. Unlike Poole et al. (2023) that computes the normal from NeRF density gradients, we compute patch-wise normal maps from the rendered depth maps, which we find to be faster and more robust. For mesh rendering, we obtain the surface color by querying the same InstantNGP neural field used in NeRF. For both NeRF and mesh, Lambertian shading is applied in the linear color space prior to tonemapping, with random point lights assigned to their respective views.

**RGBA losses.** For both NeRF and mesh, we employ RGB and alpha rendering losses to optimize the 3D parameters so that the rendered views $\tilde{x}_t$ match the intermediate denoised views $\hat{x}'_t$. For RGB, we employ a combination of pixel-wise L1 loss and patch-wise LPIPS loss (Zhang et al., 2018). For alpha, we predict the target alpha channel from $\hat{x}'_t$ using an off-the-shelf background removal network (Lee et al., 2022) as in Magic123 (Qian et al., 2024). Additionally, we soften the predicted alpha map using Gaussian blur to prevent NeRF from overfitting the initialization.

**Normal losses.** To avoid bumpy surfaces, we apply an L1.5 total variation (TV) regularization loss on the rendered normal maps:

$$\mathcal{L}_{\mathrm{N}} = \sum_{chw} \left\| w_{hw} \cdot \nabla_{hw} n_{chw}^{\mathrm{rend}} \right\|^{1.5},  \tag{10}$$

where $n_{chw}^{\mathrm{rend}} \in \mathbb{R}$ denotes the value of the $C \times H \times W$ normal map at index $(c, h, w)$, $\nabla_{hw} n_{chw}^{\mathrm{rend}} \in \mathbb{R}^2$ is the gradient of the normal map w.r.t. $(h, w)$, and $w_{hw} \in [0, 1]$ is the value of a foreground mask with edge erosion.

**Ray entropy loss for NeRF.** To mitigate fuzzy NeRF geometry, we propose a novel ray entropy loss based on the probability of sample contribution. Unlike previous works (Kim et al., 2022; Metzer et al., 2023) that compute the entropy of opacity distribution or alpha map, we consider the ray density function:

$$p(\tau) = T(\tau)\sigma(\tau),  \tag{11}$$

where $\tau$ denotes the distance, $\sigma(\tau)$ is the volumetric density and $T(\tau) = \exp - \int_0^s \sigma(\tau)d\tau$ is the ray transmittance. The integral of $p(\tau)$ equals the alpha value of the pixel, i.e., $a = \int_0^{+\inf} p(\tau)d\tau$, which is less than 1. Therefore, the background probability is $1 - a$ and a corresponding correction term needs to be added when computing the continuous entropy of the ray as the loss function:

$$\mathcal{L}_{\mathrm{ray}} = \sum_r \int_0^{+\inf} -p_r(\tau)\log p_r(\tau)d\tau - \underbrace{(1 - a_r)\log\frac{1 - a_r}{d}}_{\text{background correction}},  \tag{12}$$

where $r$ is the ray index, and $d$ is a user-defined "thickness" of an imaginative background shell, which can be adjusted to balance foreground-to-background ratio.

**Mesh smoothing losses** As per common practice, we employ the Laplacian smoothing loss (Sorkine et al., 2004) and normal consistency loss to further regularize the mesh extracted from DMTet.

**Implementation details** The weighted sum of the aforementioned loss functions is utilized to optimize the 3D representation. At each denoising step, we carry forward the 3D representation from the previous step and perform additional iterations of Adam (Kingma & Ba, 2015) optimization. During the denoising sampling process, the rendering resolution progressively increases from 128 to 256, and finally to 512 when NeRF is converted into a mesh (for texture generation the resolution is consistently 512). When the rendering resolution is lower than the diffusion resolution 512, we employ RealESRGAN-small (Wang et al., 2021b) for efficient super-resolution.

# D   DETAILS ON THE MEAN DEPTH DISTORTION (MDD) METRIC

The MDD metric is inspired by the depth distortion loss in Yu et al. (2024), which proves effective in removing floaters and improving the geometry quality. The depth distortion loss of a pixel is defined as:

$$\mathcal{L}_\mathrm{D} = \sum_{m,n} \omega_m \omega_n |\tau_m - \tau_n|, \tag{13}$$

where $m, n$ index over Gaussians contributing to the ray, $\omega_m$ is the blending weight of the $m$-th Gaussian and $\tau_m$ is the distance of the intersection point.

To compute the mean depth distortion of a view, we take the sum of depth distortion losses across all pixels and divide it by the sum of alpha values across all pixels:

$$MDD = \frac{\sum_r \mathcal{L}_{\mathrm{D}r}}{\sum_r a_r}, \tag{14}$$

where $r$ is the pixel index.

# E   MORE RESULTS

We present more qualitative comparisons in Fig. 8, 10,  11, 12, 13, 14, 15.

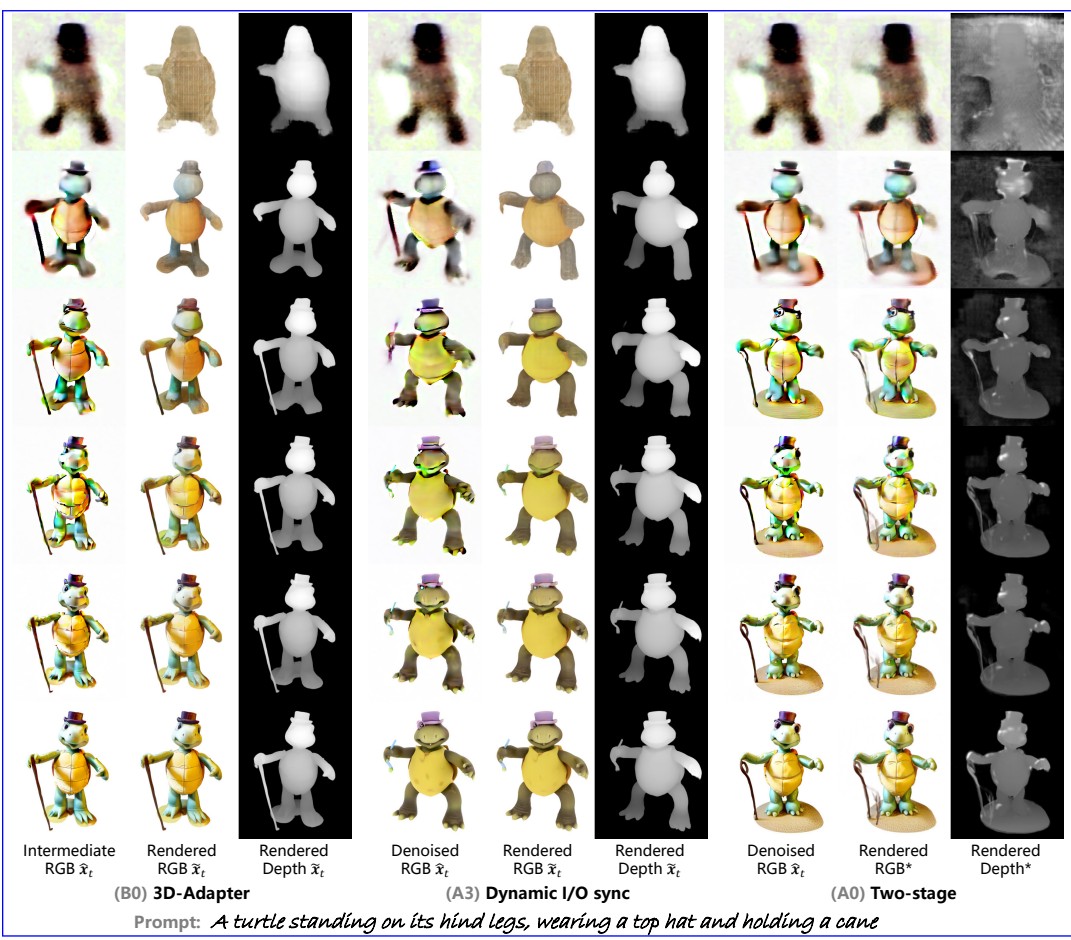

Figure 8: Text-to-3D: visualization of the multi-step sampling process. *For the two-stage method, the rendered RGB and depth maps (using the original GRM reconstructor before finetuning) are NOT a part of the sampling process, and are presented here solely for visualization.

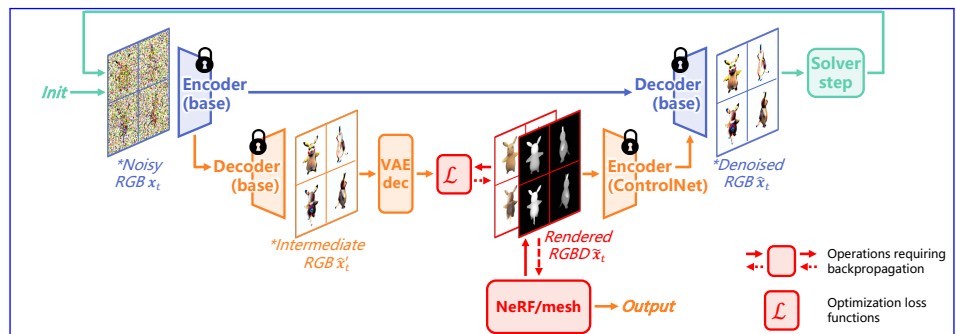

Figure 9: High-level architecture of the optimization-based 3D-Adapter. For each denoising step, the 3D representation (NeRF or mesh) is optimized to match the rendered RGB $\tilde{x}_t^{\text{RGB}}$ to the the decoded intermediate RGB $\hat{x}_t'$. The rendered RGBD maps $\tilde{x}_t$ are then fed to the ControlNet for feedback augmentation. Dense views ($\geq 32$) are typically required, although 4 views are illustrated.

Figure 10: Text-to-3D: qualitative results from the parameter sweep on $\lambda_{\text{aug}}$ and the ablation studies.

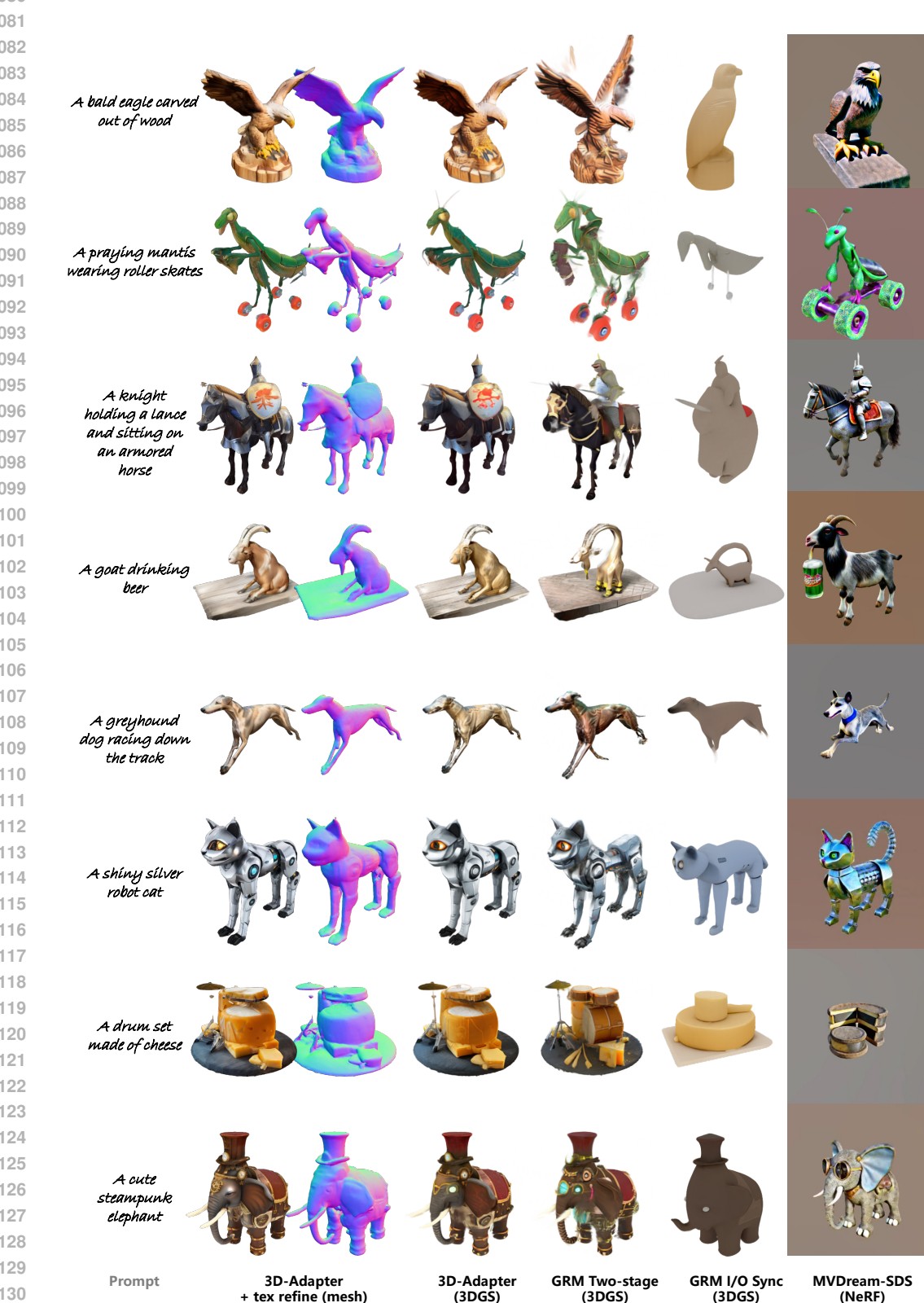

Figure 11: More comparisons on text-to-3D generation (part 1).

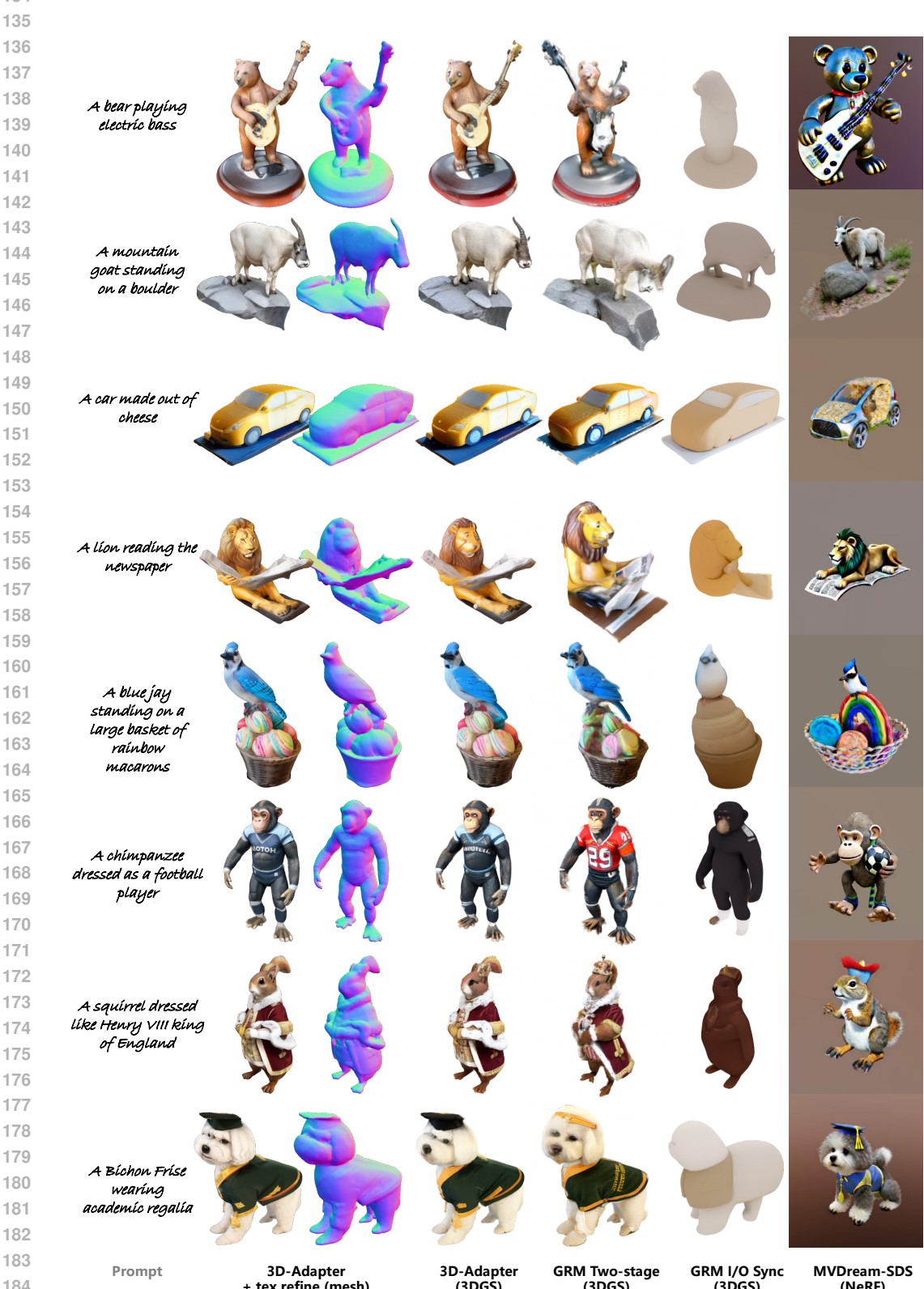

Figure 12: More comparisons on text-to-3D generation (part 2).

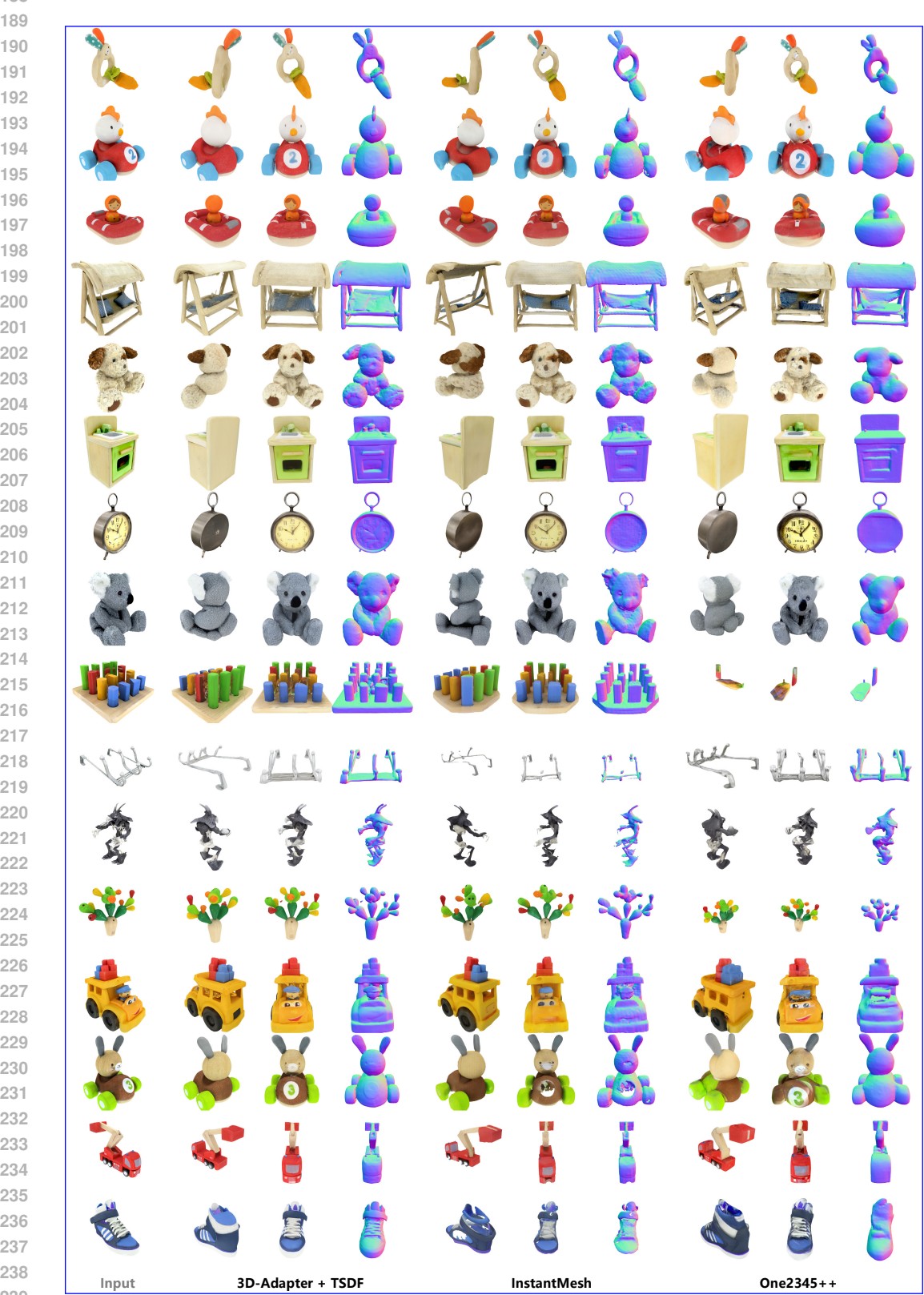

Figure 13: More comparisons on image-to-3D generation.

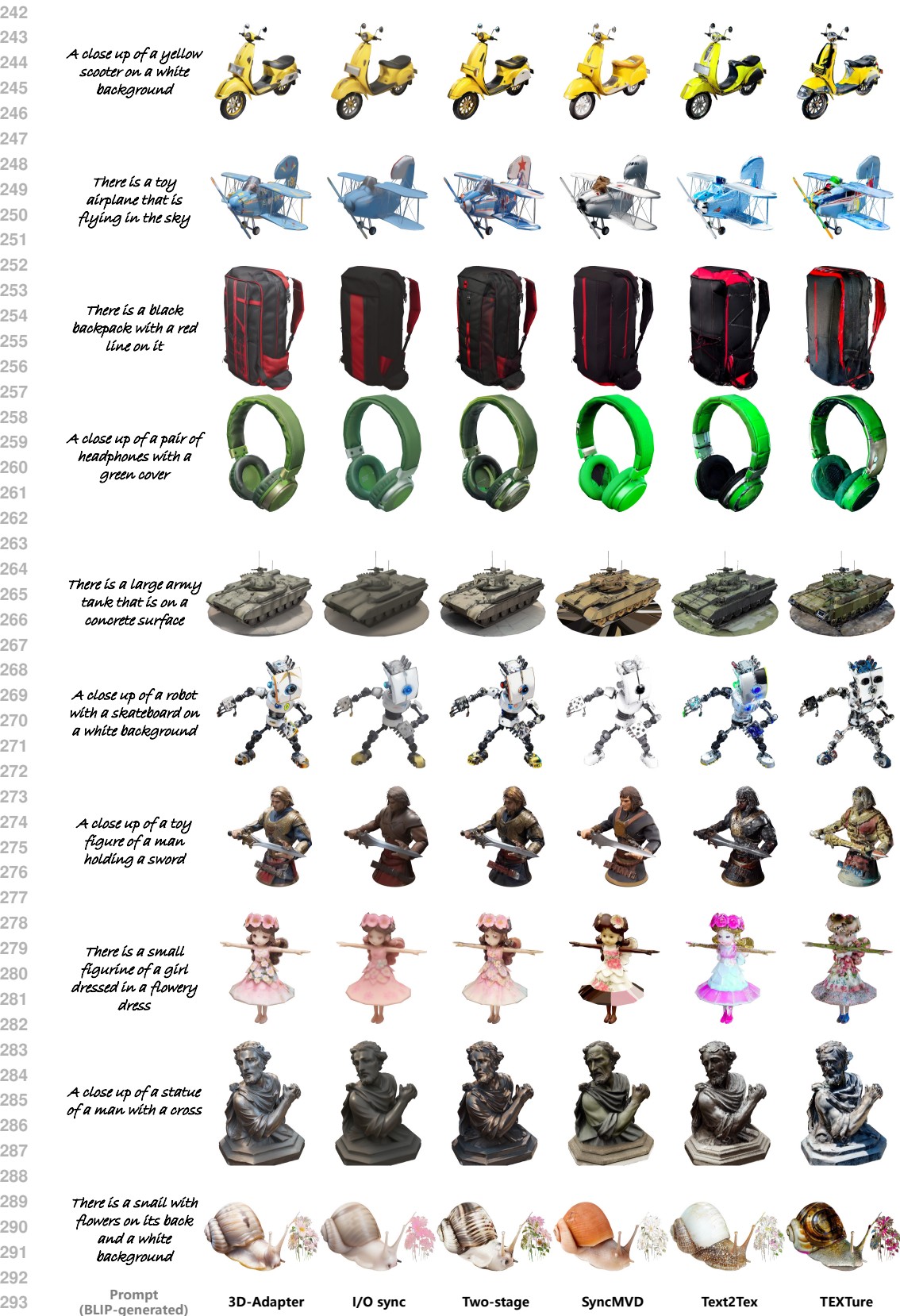

Figure 14: More comparisons on text-to-texture generation.

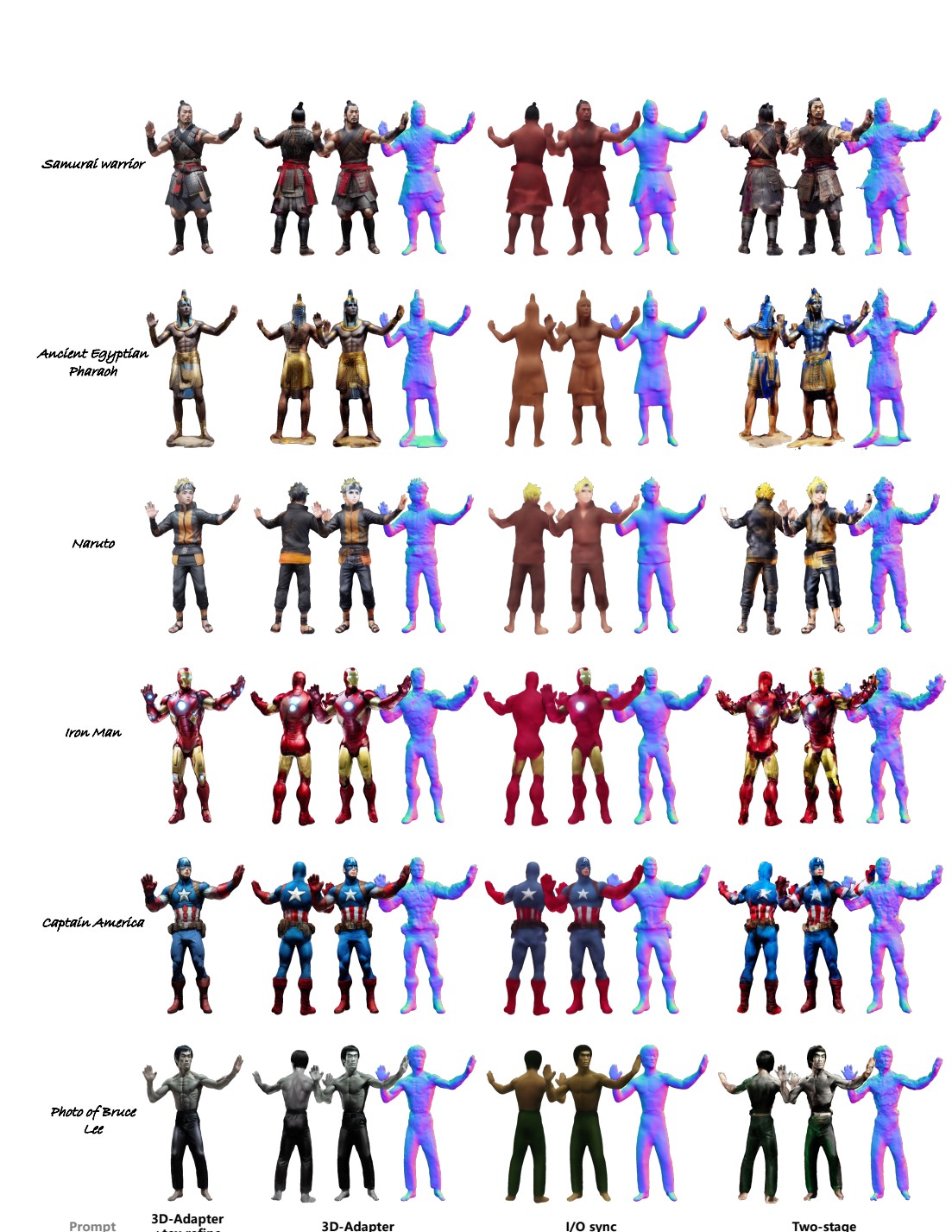

Figure 15: More comparisons on text-to-avatar generation.

