# OpenReview forum: "3D-Adapter: Geometry-Consistent Multi-View Diffusion for High-Quality 3D Generation"
_ICLR.cc/2025/Conference — Submitted to ICLR 2025_

### Official Review · Reviewer_TS5P · 2024-10-25

**Soundness:** 3
**Presentation:** 2
**Contribution:** 3
**Rating:** 6
**Confidence:** 5

**Summary:**

The paper introduces a novel plug-in module, 3D-Adapter, which enhances multi-view diffusion models to improve the quality of 3D geometry generation. By integrating 3D feedback augmentation, it infuses 3D geometry awareness into pretrained image diffusion models. The 3D-Adapter operates in two main variants: a fast feed-forward version and a flexible training-free version using neural fields and meshes. The proposed method addresses the limitations of previous two-stage approaches by maintaining the original network topology and augmenting the base model through feature addition. Extensive experiments show improvements across various tasks, including text-to-3D, image-to-3D, text-to-texture, and text-to-avatar generation.

**Strengths:**

1. The 3D feedback augmentation approach and the architecture's integration with diffusion models present a novel contribution that extends beyond typical 2D-to-3D adaptation techniques.
2. The figures effectively demonstrate the qualitative improvements of the proposed approach, particularly in challenging cases.
3. The broad applicability of 3D-Adapter to text-to-3D, image-to-3D, and text-to-texture tasks indicates potential for further research and practical applications.

**Weaknesses:**

1. **Claims About Existing Methods**: The assertion (L046-L053) that I/O sync methods degrade residual connections and cause texture quality issues is confusing and not adequately supported. More specific explanations with these methods are necessary.
2. **Inadequate Definitions**: The paper lacks clarity on the a) components of the 3D-Adapter (I understand that it includes VAE Decoder, ControlNet, and 3D Reconstruction Model) and b) the “I/O sync” baselines compared in Fig.1. Explicitly listing the modules included and the compared methods or baseline settings would aid understanding.
3. **Comparison Gaps**: The image-to-3D generation experiments do not include comparisons with CRM[1], SV3D[2], InstantMesh[3]. The text-to-texture experiments do not include metrics like FID, KID or comparisons with some established methods such as Paint3D[4] and FlashTex[5].
4. **Limitation Discussion**: Although the paper mentions the concerns about inference efficiency and shows the metric in the Appendix, more discussions about training efficiency would strengthen the paper, because it requires fine-tuning reconstruction model while keeping the multi-view diffusion model in memory.
5. **Insufficient References and Discussion of Prior Work**: a) The paper lacks citations for relevant multi-view generation methods such as Free3D[6], EpiDiff[7], and SPAD[8]. b) Additionally, it does not discuss highly relevant works like IM3D[9] and Carve3D[10]. c) While Ouroboros3D[11] and Cycle3D[12] may be considered concurrent work, it would still be valuable to include a discussion, as there are differences between these methods and the 3D-Adapter that merit discussion.

[1] CRM: Single Image to 3D Textured Mesh with Convolutional Reconstruction Model

[2] SV3D: Novel Multi-view Synthesis and 3D Generation from a Single Image using Latent Video Diffusion

[3] InstantMesh: Efficient 3D Mesh Generation from a Single Image with Sparse-view Large Reconstruction Models

[4] Paint3D: Paint Anything 3D with Lighting-Less Texture Diffusion Models

[5] FlashTex: Fast Relightable Mesh Texturing with LightControlNet

[6] Free3D: Consistent Novel View Synthesis without 3D Representation

[7] EpiDiff: Enhancing Multi-View Synthesis via Localized Epipolar-Constrained Diffusion

[8] SPAD : Spatially Aware Multiview Diffusers

[9] IM-3D: Iterative Multiview Diffusion and Reconstruction for High-Quality 3D Generation

[10] Carve3D: Improving Multi-view Reconstruction Consistency for Diffusion Models with RL Finetuning

[11] Ouroboros3D: Image-to-3D Generation via 3D-aware Recursive Diffusion

[12] Cycle3D: High-quality and Consistent Image-to-3D Generation via Generation-Reconstruction Cycle

**Questions:**

1. Could the authors elaborate on how 3D-Adapter's 3D feedback augmentation differs fundamentally from existing I/O sync techniques? Specifically, in the claim made around L051, the paper categorizes methods such as Nerfdiff, DMV3D, and VideoMV under synchronizing the denoised outputs, suggesting that these approaches disrupt residual connections and result in poor texture quality. This assertion is somewhat confusing and not evidently supported. For example, DMV3D uses a 3D reconstruction model as a multi-view denoiser, and VideoMV employs a stage-wise re-sampling strategy to fuse 3D reconstruction information. It is unclear how these methods would interfere with the residual connections in the network design. The claim requires further justification or supporting evidence to be convincing.
2. Additionally, it is recommended that the authors consider showing intermediate results during the denoising process, such as outputs from the 3D reconstruction model at various stages. This could help readers better understand the contributions of the proposed method.

---

> ### Author Response · Authors · 2024-11-23
>
> Thank you for the constructive comments. We have uploaded a revised PDF and the following are comments in response to the weaknesses.
>
> > **Claims About Existing Methods: The assertion (L046-L053) that 1/0 sync methods degrade residual connections and cause texture quality issues is confusing and not adequately supported. More specific explanations with these methods are necessary.**
>
> (Edited) We acknowledge that our use of “residual connections” may be confusing, so we have revised the text for better clarity:
>
> > Diffusion model sampling is sensitive to error accumulations (Li & van der Schaar, 2024). I/O sync methods insert 3D reconstruction and rendering operations into the denoiser in a way that disrupts the original model topology and introduces errors during each denoising step (unless reconstruction and rendering are perfect).
>
> > **Inadequate Definitions: The paper lacks clarity on the a) components of the 3D-Adapter (I understand that it includes VAE Decoder, ControlNet, and 3D Reconstruction Model) and b) the "I/O sync" baselines compared in Fig.1. Explicitly listing the modules included and the compared methods or baseline settings would aid understanding.**
>
> Edit: Your understanding is correct. In the revised PDF, we updated Figure 2 to include VAE encoders and decoders.
>
> > **Comparison Gaps: The image-to-3D generation experiments do not include comparisons with CRM[1], SV3D[2], InstantMesh[3].**
>
> Thank you for the suggestions. We have added CRM and InstantMesh to the comparisons. Notably, the strongest methods (3D-Adapter, GRM, InstantMesh, One2345++) all use Zero123++ as their base model. To the best of our knowledge, SV3D is a comparatively weaker base model.
>
> > **The text-to-texture experiments do not include metrics like FID, KID**
>
> Computing FID and KID metrics requires a reference dataset, but none of the text-to-texture models we evaluated are trained on domain-specific data, as they adapt single-view image diffusion models for texture generation. Therefore, a standardized reference dataset does not exist. Using Objaverse for this purpose is unsuitable because (a) most Objaverse objects have poor textures, and (b) the evaluated methods do not disentangle texture from lighting, requiring non-standardized lighting choices for reference data creation.
>
> > **Comparisons with some established methods such as Paint3D[4] and FlashTex[5].**
>
> Edited: Paint3D and FlashTex aim to disentangle lighting from texture, which is beyond the scope of this work (we acknowledge this as a limitation in the revised paper).
>
> > **More discussions about training efficiency would strengthen the paper, because it requires fine-tuning reconstruction model while keeping the multiview diffusion model in memory.**
>
> Thank you for the suggestions. In the revised PDF (Section 4.1 Training details), we have added the training time and VRAM requirements:
>
> > The adapter is trained with a total batch size of 16 objects on 4 A6000 GPUs **(VRAM usage peaks at 39GB)**. In phase 1, GRM is finetuned with a small learning rate of $5\times10^{-6}$ for 2k iterations (for Instant3D, **taking 3 hours**) or 4k iterations (for Zero123++, **taking 9 hours**). In phase 2, ControlNet is finetuned with a learning rate of $1\times10^{-5}$ for 5k iterations **(taking 8 hours for Instant3D and 5 hours for Zero123++)**.
>
> We think our training cost isn’t particularly heavy, when compared to training the base diffusion model such as Instant3D and Zero123++.
>
> > **Insufficient References and Discussion of Prior Work.**
>
> Edited: Thank you for providing the suggested references. All the references have been added.
>
> Remarks on some of the related work:
>
> * Ouroboros3D is a concurrent work and the high-level design is very close to our 3D-Adapter. The key difference is that Ouroboros3D feeds the rendering to the next denoising timestep, while ours operates within the current timestep. In fact, we have tested the other design as an early iteration of 3D-Adapter, and the results were slightly weaker on the text-to-texture benchmark (CLIP=26.12, Aesthetic=4.83, vs our CLIP=26.40, Aesthetic=4.85).
> * Free3D is a novel view model, its role is the same as Instant3D or Zero123++. EpiDiff, SPAD also focus on improving novel view models through epipolar attention. These work did not evaluate the rendered views from 3D representations. In contrast, our work focuses on imposing explicit 3D reconstruction and rendering. In response to Reviwer dfQt, we have tested the official code of EpiDiff, and the generated views exhibit poor consistency.
> * Cycle3D is an I/O sync method for image-to-3D. It also produces slightly blurry appearances on the backside of objects.
> * IM3D is two-stage generation but with repeated SDEdit-like refinements to the rendered views, which is an orthogonal contribution to our 3D-Adapter.
> * Paint3D and FlashTex focuses on PBR texture generation, which is beyond the scope of this work (we acknowledge this as a limitation in the revised paper).

---

> ### Author Response · Authors · 2024-11-23
>
> Comments on the questions:
>
> > **Could the authors elaborate on how 3D-Adapter's 3D feedback augmentation differs fundamentally from existing I/O sync techniques?**
>
> As stated in the paper,
>
> > We broadly define I/O sync as inserting a 3D representation and a render-ing/projecting operation at the input or output end of the denoising network to synchronize multiple views.
>
> Typical output synchronization models use rendered views as outputs for the current denoising step, fed into the diffusion solver. These models often face error accumulation (unless reconstruction and rendering are perfect) or mode collapse (discussed in Appendix A.1). For example, DMV3D is a native I/O sync model, it does not suffer from mode collapse because it is trained to predict $\hat{\mathbf{x}}_t$ as the mean of the distribution $p(\mathbf{x}_0|\mathbf{x}_t)$, where $p(\mathbf{x}_0|\mathbf{x}_t)$ is a joint probability of multiple views. However, it still suffers from error accumulation in late denoising stages (confirmed with the DMV3D authors). On the other hand, SyncMVD, an adapted I/O sync model using Stable Diffusion for texture generation, lacks domain-specific training and suffers from mode collapse due to score averaging.
>
> What makes 3D-Adapter *unique* is its ability to introduce 3D-awareness while keeping the base model's topology intact. By leveraging ControlNet’s zero-initialized weights, 3D-Adapter integrates 3D priors with minimal impact on the base model, achieving enhanced performance without quality degradation.
>
> > **Additionally, it is recommended that the authors consider showing intermediate results during the denoising process, such as outputs from the 3D reconstruction model at various stages. This could help readers better understand the contributions of the proposed method.**
>
> Thank you for the suggestion. We have added visualizations of the intermediate results in the revised PDF (Fig. 8 in the Appendix).

---

### Official Review · Reviewer_dfQt · 2024-11-03

**Soundness:** 4
**Presentation:** 4
**Contribution:** 4
**Rating:** 6
**Confidence:** 4

**Summary:**

This submission introduces 3D-Adapter, a novel module to address challenges in 3D geometry consistency synchronization in multi-view image diffusion models.
This submission identifies the fundamental limitation of existing synchronization methods working on input and output domains. Then, the authors propose to add a synchronization mechanism to the intermediate feature level (like ControlNet) to encourage 3D consistency in existing multi-view image diffusion models, instead of input and output levels.
The key idea of the mechanism is that, at each denoising step, intermediate latent representations are decoded, 3D-reconstructed, and projected back to 2D representations (RGBD), where improved 3D consistency is re-encoded, called 3D feedback augmentation.

The study explores two variants of the 3D-Adapter: a feed-forward version using a Gaussian Splatting-based model (GRM) with fine-tuning and a training-free version utilizing neural fields and meshes depending on application scenarios. The extensive experiments across text-to-3D, image-to-3D, text-to-texture, and text-to-avatar tasks demonstrate that 3D-Adapter improves generation quality in existing pre-trained models like Instant3D and Zero123++.


Overall, the submission is deemed to be well-prepared and positioned.

The proposed method itself would not be very innovative, but this reviewer found that the motivation and its theory behind the motivation are interesting. Although the researchers have been empirically aware of the limitation of the I/O sync method, the authors clearly presented the gap, which grounds the motivation of the proposed approach well.

However, some weaknesses remain that may improve the submission further.

**Strengths:**

- Clear contributions and positioning among existing research
- Clear demonstration of the limitation of I/O sync
- Resource-efficient design (training data efficient and training resource-efficient)
- Demonstration of various applications
- Noticeable improvements in terms of geometry and visual fidelity
- Clarity and presentation. The paper is well-organized and clearly presented

**Weaknesses:**

- **Lack of information on computation and memory overhead for competing methods**

  Although the authors briefly address computaitonal overhead in "Text-to-Texture Generation" and "Text-to-Avatar Generation" applications, the authors do not explicitly compare overheads in "Text-to-3D" and "Image-to-3D" cases. While it is not neccessary to win every competing method, providing these comparisons would help readers better understand the position of this work in terms of computation demands.

- **Missing related work**

  The paper is well-position among the existing work, but some recent works are missing. Since this field is very competitive and timely, it would be beneficial to acknowledge recent developments as well so that readers gains a clearer understanding of the current landscape and to discern similarities and differences.

  [C1] EpiDiff: Enhancing Multi-View Synthesis via Localized Epipolar-Constrained Diffusion, CVPR 2024

  [C2] LRM: Large Reconstruction Model for Single Image to 3D, ICLR 2024

- **Lack of comparison with recent methods in Sec. 5.4 "Text-to-Texture Generation"**

  Texture generation researches have been developed to enhance multi-view consistency quite well. A few samples of the following works are not concurrent but the past researches. The authors missed these very relevant work and only compared with past researches.

  [C3] DreamMat: High-quality PBR Material Generation with Geometry- and Light-aware Diffusion Models, SIGGRAPH 2024

  [C4] TexPainter: Generative Mesh Texturing with Multi-view Consistency, SIGGRAPH 2024

  [C5] Paint-it: Text-to-Texture Synthesis via Deep Convolutional Texture Map Optimization and Physically-Based Rendering, CVPR2024

  [C6] Paint3D: Paint Anything 3D with Lighting-Less Texture Diffusion Models, CVPR2024

  In addition, for the Text-to-3D generation, [C1] released their code, but the authors did not compare with this closely related work.


- **Confusing notation of the proposed method in the tables**

  3D-Adaptor is not a standalone model. The notation "3D-Adaptor" could mislead readers to perceive it as a standalone model. The authors are recommended to change the notations of the proposed method across all the parts: e.g., "3D-Adaptor (ours)" => "Instant3D + 3D-Adaptor (ours)" to clarify the integration.

**Questions:**

- The analysis in Appendix A helps to clarify the motivation of this work. On the other hand, the result is something that has been known in the community. Have any similar results been discussed previously in different domains? It would be convincing to add references to discuss consistency across different works in different applications.

- How much training data should be used to train GRM? As stated in the training details of Sec. 4.1, the authors fine-tuned 2000 iterations with 16 objects in a single batch, i.e., 32000 objects. Considering that a single object yields many views, it seems to be a lot for fine-tuning. What is the minimal training data to make the proposed method work? This question can also be rephrased as why the authors picked 2000 and 4000 iterations for the Instant3D and Zero123++ cases, respectively.

- Extending analysis: The authors showed why I/O sync may be bad, which is interesting. Then, another question naturally arises: why ControlNet-like feature addition (feedback augmentation) used in this work is effective? It is more interesting because the proposed method also does not guarantee a reduction of the gap in Eq. (7).

- Lines 286-288: The authors mentioned two ControlNets for superresolution and depth. Does it mean the authors use both ControlNet or is the depth ControlNet replaced with the super-resolution ControlNet? Then, how can we feed depth rendering to the superresolution ControlNet encoder without training?

- Line 287: Please provide the URL for the superresolution ControlNet as a footnote to give proper credit and to enhance the reproducibility.

- Line 485 - "texture field optimization": What is texture field optimization referred to? Is the texture field optimization applicable to the other competing methods, TEXTure, Text2Tex, and SyncMVD, in Table 6? Was it applied to them already?

**Details Of Ethics Concerns:**

There is no explicit ethical concern regarding the method itself.
However, there is a shared potential ethics concern around responsible use as with any generative model producing realistic 3D content.
The method could be misused in unconsented content reproduction and violating copyrights.

The authors are encouraged to include a paragraph of Ethics Statement (at the end of the main text before references) to address potential concerns as instructed in the ICLR author guide. A discussion of the potential implications could enhance the paper’s contribution to responsible AI practices.

---

> ### Author Response · Authors · 2024-11-23
>
> Thank you for the comments. We have uploaded a revised PDF and the following are comments in response to the weaknesses.
>
> > **Lack of information on computation and memory overhead for competing methods**
>
> Our evaluation focuses on comparing 3D-Adapter with 2-stage and I/O sync methods. Detailed runtime analysis is provided in Table 7 (Appendix) and briefly mentioned in Section 5.2. Table 7 shows that 3D-Adapter's per-step time is 0.707 sec, with 0.531 sec spent on added modules. The remaining 0.176 sec is the time for 2-stage method (excluding the second reconstruction stage). Results for the Zero123++ model are similar.
>
> Edit: In the revised PDF, we added inference time comparisons with SOTAs. Still, we need to point out that these system-level runtime comparisons are less significant due to variations in model choices and implementations.
>
> > **Missing related work**
>
> Edit: Thank you for the suggestions. We have added these papers into related work.
>
> > **Lack of comparison with recent methods in Sec. 5.4 "Text-to-Texture Generation"**
>
> DreamMat, Paint-it, Paint3D focuses on PBR texture and lighting disentanglement, which is on a different track to our 3D-Adapter. TexPainter is another I/O sync method. We have added results of TexPainter in the revised PDF.
>
> > **In addition, for the Text-to-3D generation, [C1] released their code, but the authors did not compare with this closely related work.**
>
> Edit: EpiDiff is an image-to-3D model. We have tried the official code, but our 48GB A6000 GPU failed to run the inference, as the author suggested using an 80GB GPU. The core contribution of EpiDiff is improving the network architecture using epipolar attention, while our work focus on leveraging explicit 3D reconstruction and rendering, which is on a different track to EpiDiff.
>
> > **Confusing notation of the proposed method in the tables**
>
> Many other methods are also combinations of existing multi-view diffusion models and reconstruction methods without explicitly naming them. Detailing all such methods equally would be too verbose given the space constraints.

---

> > ### Comment · Reviewer_dfQt · 2024-11-27
> >
> > Thank you for preparing the responses and the revision.
> > Some of the responses clarify my questions and concerns.
> > This reviewer thinks that this work at least contains valuable content.
> > However, this reviewer is unsatisfactory to some others of the responses and feels a bit hand-wavy (some of them are specified below).
> >
> > - > We may consider adding more baselines before the final revision, but this is time-intensive and not our priority. The paper's goal is not to compare system-level SOTAs. Instead, we aim to demonstrate improvements over 2-stage and I/O sync using the same base diffusion model and reconstruction method,
> >
> >   Disagree. [C1] is a very closely related method about a way to impose multi-view consistency, which is exactly the same focus of this submission. Thus, mentioning that it is not the authors' priority is irresponsible.
> >
> > - > DreamMat, Paint-it, Paint3D focuses on PBR texture and lighting disentanglement, which is on a different track to our 3D-Adapter.
> >
> >   Disagree. Since the key contribution of this work lies in proposing a method to enforce multi-view consistency, it is crucial to evaluate the method’s performance beyond the diffuse material regime, particularly for texture generation tasks. Limiting the scope to diffuse materials seems restrictive given current advancements, which already demonstrate that PBR or BRDF photometric properties can be obtained effortlessly from pre-trained image diffusion models. Thus, as long as the authors want to truely evaluate the multi-view consistency effect, non-diffusion material cases also should have been included for comparison.
> >
> >
> >
> > This reviewer also understands that, given the limited rebuttal time, it might be hard to reflect these comparisons.
> > On the other hand, these missing comparisons could be considered critical or not, depending on the reviewers (as Reviewer `TS5P` pointed out).
> > I'd like to hear more opinions from the other reviewers.

---

> ### Author Response · Authors · 2024-11-23
>
> Comments on the questions:
>
> > **Have any similar results been discussed previously in different domains? It would be convincing to add references to discuss consistency across different works in different applications.**
>
> To our knowledge, few application-focused papers explore the theory behind linear combinations of score functions. We have added a citation to a concurrent work (Bradley & Nakkiran, Classifier-Free Guidance is a Predictor-Corrector), which examines similar phenomena in a different context.
>
> > **How much training data should be used to train GRM? As stated in the training details of Sec. 4.1, the authors fine-tuned 2000 iterations with 16 objects in a single batch, i.e., 32000 objects. Considering that a single object yields many views, it seems to be a lot for fine-tuning. What is the minimal training data to make the proposed method work? This question can also be rephrased as why the authors picked 2000 and 4000 iterations for the Instant3D and Zero123++ cases, respectively.**
>
> The iteration numbers are set to ensure GRM performance saturates under the given schedules. GRM for Zero123++ requires more iterations because the released GRM only supports 4 views with a white background, while Zero123++ generates 6 views with a gray background, creating a larger domain gap. Using more iterations helps address this gap. Dataset sizes are detailed in Section 4.1: 47k for Instant3D and 80k for Zero123++. While we have not ablated dataset size, these numbers should have exceeded the minimal requirements.
>
> > **Extending analysis: The authors showed why I/O sync may be bad, which is interesting. Then, another question naturally arises: why ControlNet-like feature addition (feedback augmentation) used in this work is effective? It is more interesting because the proposed method also does not guarantee a reduction of the gap in Eq. (7).**
>
> Thank you for the question. For 3D-Adapters with finetuned ControlNets, the diffusion loss forces the output $\hat{\mathbf{x}}_t$ to match the mean of the distribution $p(\mathbf{x}_0|\mathbf{x}_t)$, where $p(\mathbf{x}_0|\mathbf{x}_t)$ is a joint probability of multiple views. Without 3D feedback augmentation and ControlNet, it is more difficult for the model to learn the multi-view correlations in $p(\mathbf{x}_0|\mathbf{x}_t)$, while a 3D-aware branch makes it easier to do so. For 3D-Adapter with off-the-shelf tile and depth ControlNets, we empirically observe good results (since the tile ControlNet is robust to blurry conditions), although the accuracy has no perfect guarantee.
>
> > **Lines 286-288: The authors mentioned two ControlNets for superresolution and depth. Does it mean the authors use both ControlNet or is the depth ControlNet replaced with the super-resolution ControlNet? Then, how can we feed depth rendering to the superresolution ControlNet encoder without training?**
>
> We use both ControlNets. The tile superresolution ControlNet takes the rendered RGB as input, while the depth ControlNet takes the rendered depth as input.
>
> > **Line 287: Please provide the URL for the superresolution Control Net as a footnote to give proper credit and to enhance the reproducibility.**
>
> The tile ControlNet is a part of the official ControlNet repository from the ControlNet authors.
>
> > **Line 485 - "texture field optimization": What is texture field optimization referred to? Is the texture field optimization applicable to the other competing methods, TEXTure, Text2Tex, and SyncMVD, in Table 6? Was it applied to them already?**
>
> After the final denoising step, occlusion may leave some object surfaces untextured when using texture backprojection. To address this, we use InstantNGP as a volumetric RGB representation, enabling smooth interpolation for unseen regions by optimizing it to match the final denoised views. TexPainter already employs this approach and SyncMVD could potentially adopt it. We don’t think this isthe primary factor affecting quality (TexPainter is clearly worse than SyncMVD and our I/O sync baseline due to other implementation reasons).

---

> ### Author Response · Authors · 2024-11-27
>
> Thank you for the reply. The comments you read are outdated and have been updated. Please check the latest version on openreview.
>
> Regarding Epidiff:
>
> > EpiDiff is an image-to-3D model. We have tried the official code, but our 48GB A6000 GPU failed to run the inference, as the author suggested using an 80GB GPU. The core contribution of EpiDiff is improving the network architecture using epipolar attention, while our work focus on leveraging explicit 3D reconstruction and rendering, which is on a different track to EpiDiff.
>
> We have added CRM and InstantMesh as suggested by Reviewer TS5P. Note that the best methods we have compared with (One2345++, InstantMesh, GRM) all use Zero123++ as the multi-view model, stressing the importance of the base model. EpiDiff adopts a entirely new base model, which introduces lots of uncertainties. For text-to-3D, we will further add 3DTopia for comparison.
>
> PBR texture generation is a non-trivial task. While it is entirely feasible to use pretrained diffusion models to generate PBR materials, this requires careful handling of PBR differentiable renderers and lighting. Therefore, adapting our own method to generate PBR texture is beyond the scope of our work (which already has a very large scope). **Comparing our methods with PBR texture methods isn't really meaningful because our method (and the existing baselines we have compared) renders the lightingless albedo while the PBR methods need to render their results under specific lightings, so the condition isn't the same.**
>
> >  Thus, mentioning that it is not the authors' priority is irresponsible.
>
> We are being very responsible not to mislead readers with unfair comparisons. Even if we finally get the comparison done, the result won't indicate which one of EpiDiff and 3D-Adapter is superior, because the base model we use are very different.

---

> ### Comment · Reviewer_dfQt · 2024-11-27
>
> For the EpiDiff, well.. this reviewer strongly disagrees about `while our work focus on leveraging explicit 3D reconstruction and rendering, which is on a **different track** to EpiDiff.`
> The level and way in which multi-view consistency is introduced are comparable, where both methods propose modules conditioning multi-view consistency. Thus, closely related work.
>
> For the PBR texture, I know the limitations of the proposed method the authors just mentioned (the proposed method cannot deal with view-dependent photometric characteristics due to the requirement of 3D reconstruction from multi-view images), and that's the point I pointed out in the initial review.
> That is something that needs to be discussed as a limitation, not something that can be rebutted.
>
> Considering missing important references related to multi-view consistency, texture generation applications, and theory, the way the reviewer approaches could appear to be faithless. (While I didn't mention it intentionally, the theory of Appendix A is straightly deducible from the reference [Bradley & Nakkiran]. In this case, the authors should have cited that concurrent work at the initial submission.)

---

> > ### Author Response · Authors · 2024-11-27
> >
> > Update: The limitations on texture-lighting disentanglement and PBR texture references have been added.

---

> > ### Author Response · Authors · 2024-11-28
> >
> > **Update on EpiDiff:**
> >
> > We managed to test EpiDiff, using GRM to reconstruct a 3DGS from their generated views for a fair comparison. The quantitative results are added in the revised paper (which are weaker than many other methods we have tested).
> >
> > **Analysis of EpiDiff:**
> >
> > Despite its use of epipolar attention and nearby view aggregation, EpiDiff-generated views exhibit significant flickering, leading to poor 3D consistency. Below are visualizations of the sampled views:
> > * https://ibb.co/NWSyHZj
> > * https://ibb.co/cL82YWB
> > * https://ibb.co/L6hmkqn
> > * https://ibb.co/r2fswnN
> >
> > This inconsistency causes GRM to produce severe floaters, resulting in poor rendered images:
> > * https://ibb.co/zSy5QrT
> > * https://ibb.co/LJt6XDF
> >
> > Notably, the EpiDiff paper evaluates only novel view generations rather than rendered views from an actual 3D representation, which we believe is insufficient to demonstrate 3D consistency. In contrast, 3D-Adapter explicitly employs 3D representations and renderings as strong constraints for 3D consistency. This is why we think our approach differs fundamentally from EpiDiff.

---

> ### Author Response · Authors · 2024-11-27
>
> Update: Epidiff also does not provide a full 3D generation solution in their official codebase (https://github.com/huanngzh/EpiDiff). Only the multiview diffusion model is provided, which is insufficient to reproduce the 3D generation results. All the methods we have compared with are complete 3D generation solutions (with NeRF, GS, or Mesh representations), and the final rendered images are evaluated. Again, this shows that the focus of EpiDiff is to provide a better multi-view model, instead of building a complete 3D generation pipeline.

---

> ### Author Response · Authors · 2024-11-27
>
> We understand your concerns about the PBR texture comparison. We will address this as a limitation in Section 5.4 in our next update.
>
> Regarding [Bradley & Nakkiran], first of all, it is an **concurrent** work. Secondly, our appendix was written in April 2024, way before the release of [Bradley & Nakkiran] (Aug 2024). We have already added that reference in the revised version, but we do not understand why it is **necessary** to cite a concurrent work which focuses on CFG instead of 3D generation.
>
> We acknowledge that the initial submission has a lot of missing references, which have been added in the revised version.

---

### Official Review · Reviewer_kMh5 · 2024-11-04

**Soundness:** 2
**Presentation:** 2
**Contribution:** 2
**Rating:** 5
**Confidence:** 4

**Summary:**

The authors propose 3D-adapter, a method to improve quality and 3d-consistency of existing text-to-multiview and text-to-image models by having a branch which reconstructs an object in 3D, then applies trained controlnet on rendered rgb and depth.

**Strengths:**

* Improved quality in downstream tasks over previous state-of-the-art
* Extensive final metrics in text-to-3d evaluation

**Weaknesses:**

* The quality improvement in the main text-to-3D task is 1) marginal and 2) visuials are not convincing enough to justify complicating the training pipeline. For instance, in Fig. 1, image-to-3D visuals are comparable b/w 3D-Adapter and "two-stage pipeline"
* The paper lacks significant novelty, as the use of 3D representations for synchronization has been explored before (e.g. NerfDiff). The main contribution appears to be the placement of the adapter in the parallel branch rather than at the input/output stage of the diffusion UNet. However, this seems more like a technical choice, as is the decision to train ControlNet on intermediate outputs.
* Some visuals (e.g. Fig. 3, column 5) raise concerns about the correctness of the implementation.

**Questions:**

1. Please clarify the following:
  > However, 3D reconstruction and rendering are lossy operations that disrupt residual connections.

  Since residual connections shouldn’t be disrupted if 3D reconstruction is applied before or after the UNet denoising stage, I’m unclear why this is described as a problem. Could you explain?

2. (line 142)

> A common issue with two-stage approaches is that existing reconstruction methods, often designed for or trained under conditions of perfect consistency, lack robustness to local geometric inconsistencies. This may result in floaters and **blurry textures**.

Please provide evidence that two-stage approaches specifically suffer from blurry textures.

3. (line 198)

> assuming linearity

Could you clarify what you mean by "linearity" in this context?

4. Regarding concerns over some visuals: Could you provide a detailed explanation of the I/O-sync implementation in Tables 1 and 5? Table 5 shows a slight improvement when using I/O-sync in the "baseline" model, but in Table 1, there's a significant drop from A0 to A1. The visuals in column 5 appear broken and clearly worse than those in, for example, LGM, yet the CLIP score is significantly higher. Please double-check these evaluations.

5. Could you justify the use of ControlNet? How does it compare to using pure latent fusion?

6.
> During the sampling process, the adapter performs NeRF optimization for the first 60% of the denoising steps. It then converts the color and density fields into a texture field and DMTet mesh, respectively, to complete the remaining 40% denoising steps. All optimizations are incremental, meaning the 3D state from the previous denoising step is retained to initialize the next. As a result, only 96 optimization steps are needed per denoising step. Alternatively, for texture generation only, multiview aggregation can be achieved by backprojecting the views into UV space and blending the results according to visibility

It's quite complex and specific training process, it would be great to hear about the reasoning/ablations behind these numbers.

---

> ### Author Response · Authors · 2024-11-23
>
> Thank you for the comments. We have uploaded a revised PDF and the following are comments on the weaknesses.
>
> > **The quality improvement in the main text-to-3D task is 1) marginal and 2) visuials are not convincing enough to justify complicating the training pipeline. For instance, in Fig. 1, image-to-3D visuals are comparable b/w 3D-Adapter and "two-stage pipeline".**
>
> We revised Fig. 3 to highlight the differences between 3D-Adapter and 2-stage results. Across all 3D generation comparisons, 2-stage consistently shows more floaters and fuzzy geometries, which are fixed by 3D-Adapter. This is also reflected in the MDD metric in Table 1. In Fig. 8, we also show the differences of intermediate results.
>
> The goal of 3D-Adapter is to improve geometry consistency without sacrificing visual quality, effectively removing floaters and texture seams. Ideally, 3D-Adapter should reproduce the base model's appearance unless inconsistencies arise. **We do not expect 3D-Adapter to produce completely new results**, which should be the job of the base diffusion model and the reconstruction method.
>
> > **The paper lacks significant novelty, as the use of 3D representations for synchronization has been explored before (e.g. NerfDiff). The main contribution appears to be the placement of the adapter in the parallel branch rather than at the input/output stage of the diffusion UN et. However, this seems more like a technical choice, as is the decision to train Control Net on intermediate outputs.**
>
> Edited: We have clarified the differences between previous synchronization methods (I/O sync) and our 3D-Adapter. Our approach places 3D reconstruction in a parallel branch, preserving the base model’s topology. In the revised introduction, we clarified the error accumulation problem of I/O sync:
>
> > Diffusion model sampling is sensitive to error accumulations (Li & van der Schaar, 2024). I/O sync methods insert 3D reconstruction and rendering operations into the denoiser in a way that disrupts the original model topology and introduces errors during each denoising step (unless reconstruction and rendering are perfect).
>
> Our approach, while simple, proves effective, as shown in our experiments. I/O sync often produces blurry results, whereas 3D-Adapter consistently demonstrates better visual quality.
>
> ControlNet is an intuitive choice for our method because its zero initialization is designed to minimize the impact to the base model, and the base model’s topology remains intact. Additionally, it inherits the base model’s knowledge, requires minimal fine-tuning. Other technical choices are of course possible (e.g., we have tested T2I Adapter, which performed worse than ControlNet), especially considering the other base models such as DiTs. Nevertheless, we believe our core contribution—the parallel 3D branch—is a critical innovation.
>
> > **Some visuals (e.g. Fig. 3, column 5) raise concerns about the correctness of the implementation.**
>
> We have double checked our implementation of I/O sync and found no problems. In fact, our I/O sync baseline is exceptionally strong on the text-to-texture benchmark; for text-to-3D, its MDD metric is also the best overall. Regarding the bad visual appearance, we added the explanation in the revised Appendix A.2 (edited):
>
> > While I/O sync works reasonably on our texture generation benchmark, our text-to-3D model using I/O sync (A2 in Table 1 and Fig. 3) exhibits significant quality degradation due to mode collapse. We believe the main reasons are twofold. First, the base model Instant3D generates a very sparse set of only four views, which are hard to synchronize. Second, our fine-tuned GRM reconstructor is trained using the depth loss to suppress surface fuzziness, which has a negative impact when its sharp renderings $\tilde{\mathbf{x}}_t$ are used as diffusion output. This is because a well-trained diffusion model should actually predict blurry outputs $\hat{\mathbf{x}}_t$ in the early denoising stage as the mean of the distribution $p(\mathbf{x}_0|\mathbf{x}_t)$. Only in the late stage should $\hat{\mathbf{x}}_t$ be sharp and crisp, as shown in Fig. 8.
>
> In light of this, in the revised PDF, we added a stronger I/O sync baseline (A3) with the dynamic blending technique:
>
> > As shown in Table 1 and Fig. 3, dynamic I/O sync demonstrates significant improvements in visual quality over vanilla I/O sync.
>
> Edit: Please also note that I/O sync itself is not typically the first choice for 3D generation tasks. For example, in text-to-3D and image-to-3D tasks, most state-of-the-art methods (e.g., GRM, CRM, InstantMesh) employ two-stage approaches. I/O sync is more commonly used in texture generation tasks (e.g., SyncMVD, TexPainter), where our own I/O sync implementation outperforms others. Here, its drawbacks are thoroughly analyzed both theoretically in Appendix A.1 (where the linearity assumption holds due to linear texture blending) and empirically, as demonstrated in the experiments in Table 5.

---

> ### Author Response · Authors · 2024-11-23
>
> Comments on the questions (part 1):
>
> > **Since residual connections shouldn't be disrupted if 3D reconstruction is applied before or after the UN et denoising stage, I'm unclear why this is described as a problem. Could you explain?**
>
> (Edited) We acknowledge that our use of “residual connections” may be confusing, so we have revised the text for better clarity:
>
> > Diffusion model sampling is sensitive to error accumulations (Li & van der Schaar, 2024). I/O
> sync methods insert 3D reconstruction and rendering operations into the denoiser in a way that
> disrupts the original model topology and introduces errors during each denoising step (unless
> reconstruction and rendering are perfect).
>
> > **Please provide evidence that two-stage approaches specifically suffer from blurry textures.**
>
> Thank you for pointing this out. This is our writing issue. It should be "texture seams" rather than "blurry textures". We have revised the text:
>
> > A common issue with two-stage approaches is that existing reconstruction methods, often designed for or trained under conditions of perfect consistency, lack robustness to local geometric inconsistencies. This may result in floaters and **texture seams**.
>
> Texture seams are very noticeable in the text-to-texture and text-to-avatar results.
>
> > **Could you clarify what you mean by "linearity" in this context?**
>
> We have added a brief explanation in our revised Appendix:
>
> > When performing diffusion ODE sampling using the common Euler solver, a linear input sync operation **(e.g., linear blending or optimizing using the L2 loss)** is equivalent to...
>
> To clarify, this means that the synchronized output is a linear combination of pre-synchronization views. This assumption made to simplify theoretical analysis. While some synchronization methods (e.g., our GRM-based I/O sync) are not strictly linear, the mode collapse issue from score averaging still persists.
>
> > **Regarding concerns over some visuals: Could you provide a detailed explanation of the I/O-sync implementation in Tables 1 and 5? Table 5 shows a slight improvement when using I/O -sync in the "baseline" model, but in Table 1, there's a significant drop from AO to A1.**
>
> Table 5 focuses on texture generation. I/O sync is particularly effective for this task, since the depth ControlNet already provides strong conditioning, making view synchronization relatively easy. Notably, TexPainter and SyncMVD (shown in Table 6) are typical I/O sync methods for texture generation, though implemented differently from ours (still, our own implementation achieves better quality and efficiency).
>
> In Table 1, A1 represents the I/O sync baseline using the original GRM without finetuning. The unfinetuned GRM is incompatible with coarse intermediate outputs, and will lead to severe error accumulation and poor geometry with floaters (reflected in the high MDD metric). After finetuning the GRM (A2), the MDD metric improves substantially, even outperforming the 3D-Adapter in geometry. The drop in appearance metrics following finetuning is explained in the additional analysis provided in Appendix A2 (edited):
>
> > While I/O sync works reasonably on our texture generation benchmark, our text-to-3D model using I/O sync (A2 in Table 1 and Fig. 3) exhibits significant quality degradation due to mode collapse. We believe the main reasons are twofold. First, the base model Instant3D generates a very sparse set of only four views, which are hard to synchronize. Second, our fine-tuned GRM reconstructor is trained using the depth loss to suppress surface fuzziness, which has a negative impact when its sharp renderings $\tilde{\mathbf{x}}_t$ are used as diffusion output. This is because a well-trained diffusion model should actually predict blurry outputs $\hat{\mathbf{x}}_t$ in the early denoising stage as the mean of the distribution $p(\mathbf{x}_0|\mathbf{x}_t)$. Only in the late stage should $\hat{\mathbf{x}}_t$ be sharp and crisp, as shown in Fig. 8.
>
> In light of this, in the revised PDF, we added a stronger I/O sync baseline (A3) with the dynamic blending technique.
>
> Edit: Please also note that I/O sync itself is not typically the first choice for 3D generation tasks. For example, in text-to-3D and image-to-3D tasks, most state-of-the-art methods (e.g., GRM, CRM, InstantMesh) employ two-stage approaches. I/O sync is more commonly used in texture generation tasks (e.g., SyncMVD, TexPainter), where our own I/O sync implementation outperforms others. Here, its drawbacks are thoroughly analyzed both theoretically in Appendix A.1 (where the linearity assumption holds due to linear texture blending) and empirically, as demonstrated in the experiments in Table 5.

---

> ### Author Response · Authors · 2024-11-23
>
> Comments on the questions (part 2):
>
> > **The visuals in column 5 appear broken and clearly worse than those in, for example, LGM, yet the CLIP score is significantly higher. Please double-check these evaluations.**
>
> Column 5 shows the two-stage results from the original GRM. While some results may appear broken, more outputs are included in the appendix for reference. We observed that the CLIP score is not sensitive to broken geometry, which is why we rely on the MDD score for evaluating geometry quality. Our evaluation results in Table 2 are valid and align well with the original GRM paper.
>
> > **Could you justify the use of ControlNet? How does it compare to using pure latent fusion?**
>
> Thank you for the question. As noted above, we use ControlNet to preserve the original model topology. Since ControlNet uses zero-initialized weights, its impact to the base model is minimal, allowing 3D-awareness to be introduced without degrading quality.
>
> Latent fusion has been tested in our added dynamic I/O sync baseline, which performs dynamic fusion between the UNet outputs and the rendered outputs in the latent space. While this improves upon vanilla I/O sync, the quality remains below the 2-stage baseline and 3D-Adapter. Details are provided in the revised Appendix A.2.
>
> > **It's quite complex and specific training process, it would be great to hear about the reasoning/ablations behind these numbers.**
>
> The reconstruction method is not part of our main contributions, and we do not claim it to be optimal, though we tuned it extensively. More details are in the Appendix.
>
> The purpose of this section is to provide a consistent platform for testing the optimization-based 3D-Adapter and comparing it to 2-stage and I/O sync baselines. The text-to-avatar task effectively demonstrates 3D-Adapter's advantages over these baselines when using the same reconstructor.

---

> > ### Comment · Reviewer_kMh5 · 2024-11-26
> >
> > I'd like to thank the authors for their clarifications. After considering the issues raised by other reviewers and authors' responses, I am upgrading my ratings for "contribution" (to "fair") and my overall assessment to 5.
> > However, based on the results, I still find the contribution to be of limited significance, and have doubts about whether this setup will be built upon in the research community. To reflect the uncertainty in this judgment, I am lowering the confidence of my review to 4.

---

> ### Author Response · Authors · 2024-11-27
>
> Thank you for reconsidering your assessment! While we acknowledge the uncertainty around the adoption of new methodologies, we believe our work provides a strong foundation for further exploration:
> * **Versatility**: Our experiments span text-to-3D, image-to-3D, text-to-texture, and text-to-avatar setups, showcasing broad applicability. Other applications, such as panorama generation and 4D generation, could also benefit from improved synchronization methods like 3D-Adapter.
> * **Ease of Integration**: 3D-Adapter maintains the base model's topology, making it plug-and-play if existing ControlNets and reconstructors are available. For example, we demonstrated its use with pretrained ControlNet and customized Stable Diffusion for SOTA texture generation—no training required, ensuring accessibility for the community.
> * **Significance of Results**: Our experiments clearly demonstrate improvements across tasks, both visually and quantitatively, reinforcing the practical impact of our approach. Please note that 3D generation is a very competitive field, and our improvements in metrics are not trivial.

---

### Official Review · Reviewer_RiGq · 2024-11-04

**Soundness:** 3
**Presentation:** 3
**Contribution:** 3
**Rating:** 6
**Confidence:** 5

**Summary:**

The paper suggests a novel approach by introducing the 3D feedback augmentation adapter. It presents methods for applying this adapter to both feed-forward 3D generation and pretrained 2D models. The design incorporates a comprehensive 3D rendering process and a fine-tuning stage utilizing ControlNet to enhance the model's 3D awareness and multi-view consistency. Through experiments, the paper demonstrates the superior performance of the 3D-Adapter by applying it to various tasks and diverse base models, showcasing its effectiveness and versatility across different applications.

**Strengths:**

1. The idea behind the 3D-Adapter is conceptually novel. It incorporates a lightweight 3D rendering pipeline within the diffusion sampling process through a mechanism that balances computational efficiency and multi-view consistency.


2. While there may be limitations in terms of generalizability, the 3D-Adapter has demonstrated significant versatility through its application across various tasks and base models, as shown through comprehensive experiments.

**Weaknesses:**

1. Clarity of exposition: A more comprehensive explanation of the relationship between GRM and models such as Instant3D and Zero123++ would greatly enhance the reader's understanding. It is particularly important to elucidate whether GRM builds 3D representations from the views generated by Instant3D and Zero123++ and whether fine-tuning of GRM is involved in this process.

2, Limitations in the scalability of 3D-Adapter:
- According to Equation 2, it appears that the 3D-Adapter could potentially be applied to other I/O sync-based pretrained models. However, it is necessary to discuss whether incorporating ControlNet into I/O sync pretrained models beyond GRM would yield similar results, and why Training Phase 1 is essential. If Training Phase 1 is a crucial step for optimizing the performance of 3D-Adapter, including GRM, it should be examined whether this phase is equally necessary when applied to other models.

- Despite utilizing various 3D representation techniques and loss functions, such as NeRF and DMTet, ensuring global semantic consistency remains challenging. This inherent limitation highlights the need for additional conditions to achieve robust 3D generation. Consequently, this raises concerns about the 3D consistency of 3D-Adapter in more general text-to-3D or image-to-3D tasks outside of specialized domains, such as avatar generation.

**Questions:**

- The role of augmentation guidance appears to be significant. Could you provide visual evaluations in addition to Table 1 to better illustrate this effect?
- Please include a dedicated pipeline figure specifically for the process described in Section 4.2 (line 289) to enhance clarity.
- What are the results of using the optimization-based 3D-Adapter without additional conditioning?

---

> ### Author Response · Authors · 2024-11-23
>
> Thank you for the constructive comments! We have uploaded a revised PDF, which hopefully addresses some of the clarity issues. The following are comments in response to the weaknesses and questions.
>
> > **1. Clarity of exposition: A more comprehensive explanation of the relationship between GRM and models such as Instant3D and Zero123++ would greatly enhance the reader's understanding. It is particularly important to elucidate whether GRM builds 3D representations from the views generated by Instant3D and Zero123++ and whether fine-tuning of GRM is involved in this process.**
>
> To clarify, Instant3D or Zero123++ serves as the base model, while GRM acts as the 3D reconstructor. This relationship is outlined in Section 4 and illustrated in Figure 2(c). GRM constructs 3D representations from the base model's intermediate denoised views during both fine-tuning and inference.
>
> > **According to Equation 2, it appears that the 3D-Adapter could potentially be applied to other I/O sync-based pretrained models. However, it is necessary to discuss whether incorporating Control Net into I/O sync pretrained models beyond GRM would yield similar results.**
>
> In Table 5, we tested 3D-Adapter combined with I/O sync for texture generation (which is not based on GRM but texture backprojection). As expected, this combination underperforms compared to 3D-Adapter alone due to error accumulation and blur introduced by I/O sync (see Figure 5 and the drop in Aesthetic score).
>
> > **… and why Training Phase 1 is essential. If Training Phase 1 is a crucial step for optimizing the performance of 3D-Adapter, including GRM, it should be examined whether this phase is equally necessary when applied to other models.**
>
> Edited: Training Phase 1 involves GRM finetuning. As shown in the I/O sync experiments (A2 vs A1), only the finetuned GRM improves geometry quality (indicated by the MDD metric and visualized in Fig. 8). Additionally, C0 vs A0 also reveals the contribution of the finetuned GRM in the 2-stage setting (C0 removes the feedback and effectively becomes 2-stage). These evidences reveal that phase 1 is also beneficial to other models. Given these evidences, it is undoubted that 3D-Adapter should also use the finetuned GRM, and testing 3D-Adapter without GRM finetuning is unnecessary. However, if you're interested, we can test this setting for the rebuttal.
>
> > **Despite utilizing various 3D representation techniques and loss functions, such as NeRF and DMTet, ensuring global semantic consistency remains challenging.**
>
> This is very true and we have discussed it in the 4-th paragraph of Section 4.2. We stated in the introduction that 3D-Adapter is limited to fixing local geometry inconsistencies, and the global semantic consistency is beyond the scope of this work.
>
> > **The role of augmentation guidance appears to be significant. Could you provide visual evaluations in addition to Table 1 to better illustrate this effect?**
>
> Edited: Thank you for pointing this out. We have added visual comparisons in the revised Appendix (Fig. 10).
>
> > **Please include a dedicated pipeline figure specifically for the process described in Section 4.2 (line 289) to enhance clarity.**
>
> Thank you for the suggestion. We will add it in our next revision.
>
> > **What are the results of using the optimization-based 3D-Adapter without additional conditioning.**
>
> Thank you for the good question. This will lead to the Janus multi-face problem, since 3D-Adapter alone does not guarantee high-level global semantic consistency.

---

> ### Author Response · Authors · 2024-11-25
>
> Thank you for the timely response!
>
> Regarding the clarity issues, we wil continue to revise the paper based on the comments from other reviewers. We would appreciate it if you could provide more detailed suggestions on writing improvement.
>
> The claims about the I/O sync methods are analyzed in depth in the Appendix (also revised and updated). Please let us know if you find any part of it unclear.
>
> Edit:  We further revised the text to clarify the error accumulation issue, and added a reference to the paper *On Error Propagation of Diffusion Models*:
>
> > Diffusion model sampling is sensitive to error accumulations (Li & van der Schaar, 2024). I/O sync methods insert 3D reconstruction and rendering operations into the denoiser in a way that disrupts the original model topology and introduces errors during each denoising step (unless reconstruction and rendering are perfect).
>
> The ControlNet's effectiveness can be directly reflected by comparing B0 and C0 in table 1 (both use the same finetuned GRM). For texture and avatar generation, disabling ControlNet feedback is equivalent to the two-stage baseline and we have shown the differences. Reviewer kMh5 has asked about the comparison between ControlNet and latent fusion, and we have clarified that latent fusion is equivalent to our newly added dynamic I/O sync baseline. Please let us know if you have questions about these validations.

---

> ### Author Response · Authors · 2024-11-25
>
> Some additional comments on the I/O sync baseline:
>
> I/O sync itself is not typically the first choice for 3D generation tasks. For example, in text-to-3D and image-to-3D tasks, most state-of-the-art methods (e.g., GRM, CRM, InstantMesh) employ **two-stage** approaches. I/O sync is more commonly used in texture generation tasks (e.g., SyncMVD, TexPainter), where our own I/O sync implementation outperforms others. Furthermore, its drawbacks are thoroughly analyzed both theoretically in Appendix A.1 (where the linearity assumption holds due to linear texture blending) and empirically, as demonstrated in the experiments in Table 5.
>
> If you find our evidence insufficient, please be specific on the aspects that need improvement so we can better address them.

---

### Official Review · Reviewer_95MW · 2024-11-09

**Soundness:** 2
**Presentation:** 2
**Contribution:** 3
**Rating:** 5
**Confidence:** 4

**Summary:**

The paper introduces 3D-Adapter, a plug-in module aimed at improving 3D consistency in multi-view diffusion models for 3D generation. By integrating a combination of 3D representation and ControlNet for depth conditioning into the denoising framework, 3D-Adapter enhances 3D structure coherence across views without modifying the core model's topology / weights. The authors present two 3D-Adapter variants: a fast feed-forward method using Gaussian splatting (GRM) and a flexible, training-free approach utilizing NeRF optimization. Extensive evaluations across text-to-3D, image-to-3D, and text-to-texture tasks demonstrate that 3D-Adapter improves geometry quality and coherence over prior methods, providing a robust solution for multi-view 3D generation tasks.

**Strengths:**

By integrating depth conditioning via ControlNet, the 3D-Adapter provides a straightforward yet effective solution to incorporate 3D priors within 2D diffusion frameworks as a part of denoising process.

The paper presents extensive quantitative and qualitative evaluations across multiple configurations and baselines.

The proposed 3D-Adapter outperforms several prior methods in 3D generation quality, demonstrating improvements on widely used metrics like CLIP score and aesthetic score.

**Weaknesses:**

The paper’s scope is broad, which obscures the clarity of its main contributions and makes it challenging to pinpoint specific innovations with reusable community value. Instead of covering multiple branches, e.g., GRM vs. NeRF optimization, a more focused examination on a single GRM pipeline could provide deeper insights, making the contribution more accessible and actionable for the community. Some comparisons and insights appear irrelevant, e.g. the text-to-avatar section, as the paper primarily addresses general 3D consistency issues rather than avatar-specific issues. This creates confusion regarding the paper’s core contributions.

Introducing per-step 3D → depth → ControlNet prediction comes with computational overhead. The paper could explore strategies to mitigate this, such as reducing the frequency of applying the adapter or skipping initial steps. Since per-step VAE decoding is probably the main bottleneck, adopting a lightweight VAE alternative, e.g. TinyAE, could yield substantial speed gains.

Some design choices around the ControlNet addition appear understudied. Introducing depth ControlNet itself may add a substantial 3D prior, but this effect is not adequately considered and is instead attributed mainly to ControlNet's role in denoising. Additionally, in the GRM branch, ControlNet is trained from scratch, which seems counterintuitive and is not supported with ablation studies.

The paper was fairly difficult to read and navigate, primarily due to its broad scope and, to a lesser extent, its writing style.

**Questions:**

(1) What is the rationale behind the significant drop in metrics in Table 1 when adding IO sync and GRM finetuning to the two-stage baseline? Also GRM finetuning does not appear to be evaluated in the 3D-Adapter case.

(2) The SOTA selection in Table 2 seems debatable; could the authors clarify the choice of methods over more recent approaches?

(3) How does the output quality compare to I/O sync + tiled ControlNet, and how critical is ControlNet’s role during denoising relative to the 3D prior that could be introduced simply by depth conditioning in IO mode?

---

> ### Author Response · Authors · 2024-11-23
>
> Thank you for the comments! We have uploaded a revised PDF, which hopefully addresses some of the clarity issues. The following are comments in response to the weaknesses.
>
> > **By integrating depth conditioning via ControlNet...**
>
> Correction: 3D-Adapter employs **RGB**+depth ControlNets. In our texture generation benchmark, all baselines already use depth ControlNet; the improvement in quality comes from the additional RGB feedback provided by the 3D-Adapter.
>
> To clarify, our concept of "geometry consistency" goes beyond object surface geometry (depth-related). It involves "ensuring precise 2D–3D alignment of local features and maintaining geometric plausibility." The RGB ControlNet provides essential feedback to ensure that the texture and silhouette of the object are aligned across different views.
>
> > **The paper's scope is broad, which obscures the clarity of its main contributions and makes it challenging to pinpoint specific innovations with reusable community value.**
>
> We have revised the manuscript, which hopefully better clarifies the scope and contributions.
> > * We propose 3D-Adapter, which enables high-quality 3D generation with enhanced multi-view geometry consistency by integrating a 3D feedback module into a base image diffusion model.
> > * We demonstrate that 3D-Adapter is compatible with various base models and reconstruction methods, making it highly adaptable to a range of tasks.
> > * We conduct extensive experiments to show that 3D-Adapter improves geometry consistency while preserving visual quality, outperforming previous methods on text-to-3D, image-to-3D, and text-to-texture tasks.
>
> Again, the main contribution of this paper is the high-level design of 3D-Adapter. This is not just one specific model, and we need to demonstrate its compatibility with multiple base models and reconstruction methods to show its generalizability.
>
> > **Some comparisons and insights appear irrelevant, e.g. the text-to-avatar section, as the paper primarily addresses general 3D consistency issues rather than avatar-specific issues.**
>
> The focus of this section is not on text-to-avatar tasks (hence, we do not claim SOTA or compare with domain-specific methods) but on demonstrating the compatibility of 3D-Adapter with optimization-based 3D reconstruction methods and off-the-shelf ControlNets. Comparing 3D-Adapter with 2-stage and I/O sync baselines across various setups is critical, as different methods excel in different contexts. For example, I/O sync performs poorly in GRM-based text-to-3D setups but is competitive in optimization-based avatar generation and texture reprojection tasks, which is explained in Appendix A.2.
>
> Additionally, these setups provide valuable insights, as noted by Reviewer RiGq in their comments on optimization-based 3D-Adapters.
>
> > **Introducing per-step 3D depth Control Net prediction comes with computational overhead. The paper could explore strategies to mitigate this, such as reducing the frequency of applying the adapter or skipping initial steps. Since per-step VAE decoding is probably the main bottleneck, adopting a lightweight VAE alternative, e.g. TinyAE, could yield substantial speed gains.**
>
> This overhead is not unique to our 3D-Adapter, it also affects I/O sync methods. Acceleration strategies like skipping steps or using lightweight VAEs are orthogonal to our contributions and applicable to both 3D-Adapter and I/O sync. While they would improve efficiency, it is an engineering effort beyond the scope of our main contributions.
>
> > **Some design choices around the ControlNet addition appear understudied. Introducing depth ControlNet itself may add a substantial 3D prior, but this effect is not adequately considered and is instead attributed mainly to ControlNet's role in denoising.**
>
> Again, we use **RGB**+D ControlNets, not just depth ControlNets. The main idea of 3D-Adapter requires the rendered images to be encoded and fused into the features of the base model without disrupting the original model topology, and ControlNet is an intuitive choice since its zero initialization is designed to minimize the impact to the base model.
>
> Quantitatively, the ControlNet's effectiveness can be directly reflected by comparing B0 and C0 in table 1 (both use the same finetuned GRM). For texture and avatar generation, disabling ControlNet feedback is *equivalent* to the two-stage baseline and we have shown the differences.
>
> > **Additionally, in the GRM branch, ControlNet is trained from scratch, which seems counterintuitive and is not supported with ablation studies.**
>
> Correction: The ControlNet is not trained from scratch but fine-tuned from the base model encoder, following standard ControlNet training. This preserves most of the model's knowledge and doesn't require extensive training. Additionally, for Instant3D and Zero123++, pretrained ControlNets are unavailable. We attempted to use existing Stable Diffusion ControlNets, but they performed poorly.

---

> ### Author Response · Authors · 2024-11-23
>
> Comments on the questions:
>
> > **What is the rationale behind the significant drop in metrics in Table 1 when adding IO sync and GRM finetuning to the two-stage baseline?**
>
> After adding I/O sync (A1), the problem is that the original GRM is incompatible with coarse intermediate outputs, and will lead to severe error accumulation. After finetuning the GRM (A2), the appearance metrics becomes worse but the geometry metric MDD improves significantly. The reason is explained in Appendix A2 (edited):
>
> > While I/O sync works reasonably on our texture generation benchmark, our text-to-3D model using I/O sync (A2 in Table 1 and Fig. 3) exhibits significant quality degradation due to mode collapse. We believe the main reasons are twofold. First, the base model Instant3D generates a very sparse set of only four views, which are hard to synchronize. Second, our fine-tuned GRM reconstructor is trained using the depth loss to suppress surface fuzziness, which has a negative impact when its sharp renderings $\tilde{\mathbf{x}}_t$ are used as diffusion output. This is because a well-trained diffusion model should actually predict blurry outputs $\hat{\mathbf{x}}_t$ in the early denoising stage as the mean of the distribution $p(\mathbf{x}_0|\mathbf{x}_t)$. Only in the late stage should $\hat{\mathbf{x}}_t$ be sharp and crisp, as shown in Fig. 8.
>
> In light of this, in the revised PDF, we added a stronger I/O sync baseline (A3) with the dynamic blending technique.
>
> > As shown in Table 1 and Fig. 3, dynamic I/O sync demonstrates significant improvements in visual quality over vanilla I/O sync. Its MDD score indicates that its geometry consistency lies between vanilla I/O sync and 3D-Adapter. However, the visual quality of dynamic I/O sync is still clearly below that of the two-stage method and 3D-Adapter. While it is possible to tune a better blending weight $\lambda_t^\text{sync}$ , we believe it is very difficult to reduce the gap due to the aforementioned challenges brought by our model setup.
>
> Edited: Please also note that I/O sync itself is not typically the first choice for 3D generation tasks. For example, in text-to-3D and image-to-3D tasks, most state-of-the-art methods (e.g., GRM, CRM, InstantMesh) employ two-stage approaches. I/O sync is more commonly used in texture generation tasks (e.g., SyncMVD, TexPainter), where our own I/O sync implementation outperforms others. Here, its drawbacks are thoroughly analyzed both theoretically in Appendix A.1 (where the linearity assumption holds due to linear texture blending) and empirically, as demonstrated in the experiments in Table 5.
>
> > **Also GRM finetuning does not appear to be evaluated in the 3D-Adapter case.**
>
> Edited: 3D-Adapter uses the finetuned GRM by default. As shown in the I/O sync experiments (A2 vs A1), only the finetuned GRM improves geometry quality (indicated by the MDD metric and visualized in Fig. 8). Additionally, C0 vs A0 also reveals the contribution of the finetuned GRM in the 2-stage setting (C0 removes the feedback and effectively becomes 2-stage). Given these evidences, testing 3D-Adapter without GRM finetuning is unnecessary. However, if you're interested, we can test this setting for the rebuttal.
>
> > **The SOTA selection in Table 2 seems debatable; could the authors clarify the choice of methods over more recent approaches?**
>
> Edited: LGM and GRM are already the latest ECCV 2024 SOTAs. In the revised PDF, we also added a new comparison (3DTopia). In general, text-to-3D is a less crowded field compared to image-to-3D, thus more SOTAs are found in the image-to-3D benchmark (Table 3).
>
> > **How does the output quality compare to I/O sync + tiled ControlNet, and how critical is ControlNet's role during denoising relative to the 3D prior that could be introduced simply by depth conditioning in IO mode?**
>
> We added an experiment in Table 5 combining 3D-Adapter with I/O sync for texture generation. As expected, this combination performs worse than 3D-Adapter alone due to error accumulation and blur introduced by I/O sync (shown in Figure 5 and indicated by the drop in Aesthetic score). We tested this combination in the text-to-texture setup because I/O sync performs well here, whereas it significantly degrades text-to-3D results, making such a combination impractical. Again, this highlights the importance of testing 3D-Adapter across different setups, even if it adds complexity.

---

### Author Response · Authors · 2024-11-28
**Summary of Revisions**

**Some of our comments have been edited to reflect the latest revisions. Please refer to the latest comments on OpenReview instead of the email correspondence. Thank you for your understanding!**

We sincerely thank all the reviewers for their comments and discussions. We understand that the content of this paper is dense, and we greatly appreciate the reviewers’ efforts in carefully reading the manuscript and rebuttal and providing constructive suggestions for improvement.

We have thoroughly addressed all concerns and uploaded a fully revised manuscript. Below is a summary of the major changes:

* Improved Writing and Clarity:
  * Revised half of the introduction to better summarize this work and highlight the contributions.
  * Refined our claims about the I/O sync baseline and provided more in-depth analysis in Appendix A.
  * Revised Section 4.2 for improved clarity.
  * Updated Fig. 2 by adding VAE encoding and decoding blocks for better clarity. Additionally, Fig. 9 was added to illustrate the optimization-based 3D-Adapter.

* References:
  * Incorporated and discussed all suggested references, including concurrent work.

* New Comparisons:
  * Added comparisons with InstantMesh, CRM, TexPainter, and 3DTopia. Updated corresponding figures.
  * **Updated**: Comparison with EpiDiff is also added now, as requested by Reviewer dfQt

* New Experiment Results:
  * Table 1, Figure 3: (A3) Dynamic I/O sync
  * Table 5, Figure 5: 3D-Adapter + I/O sync
  * Figure 8: Visualization of the intermediate results
  * Figure 10: Qualitative results of Table 1 B0-C1

* Training Details:
  * Added VRAM usage and training hours in Section 4.1.

* Inference Times:
  * Reported inference times for all SOTA comparisons.

* Limitations:
  * Acknowledged that 3D-Adapter's text-to-texture pipeline does not disentangle texture from lighting.

---

### Meta-Review · Area_Chair_dYeV · 2024-12-22

**Metareview:**

The paper presents a method to incorporate depth prior into the framework of 3D generation. The paper receives mixed ratings from the reviewers. The reviewers have some concerns in terms of several perspectives. First, the reviewer argues that the novelty of the presented work is not elaborated clearly. The geometric consistency issue seems to be the key contribution that the paper targets, however, it is a bit unexpected that the depth prior is involved for 3D generation, and also, this step with diffusion will bring additional overhead as mentioned by one of the reviewers. Another issue criticized by one reviewer is that the proposed geometry consistency regularization seems to not improve the performance clearly, and also some visualizations raised potential concerns about implementation correctness. Finally, quite a few comments regarding the presentation quality were raised by the reviewers. Based on these critical comments, AC decided to reject this paper for this time.

**Additional Comments On Reviewer Discussion:**

The reviewers asked for clarification of key contributions and some important details and requested more experiments to show the effectiveness. The reviewers are not fully satisfied with the rebuttal of the authors.

---

### Decision · Program_Chairs · 2025-01-22

Reject